


# Contribution of regional aerosol nucleation to low-level CCN in an Amazonian deep convective environment: Results from a regionally nested global model

Xuemei Wang[1], Hamish Gordon[2], Daniel P. Grosvenor[1], Meinrat O. Andreae[3], and Ken S. Carslaw[1]

[1]School of Earth and Environment, University of Leeds, LS2 9JT, Leeds, United Kingdom
[2]College of Engineering, Carnegie Mellon University, PA 15213, United States
[3]Biogeochemistry, Multiphase Chemistry, and Particle Chemistry Departments, Max Planck Institute for Chemistry, Mainz, Germany

**Correspondence:** Ken S. Carslaw (k.s.carslaw@leeds.ac.uk)

**Abstract.** Global model studies and observations have shown that downward transport of aerosol nucleated in the free troposphere is a major source of cloud condensation nuclei (CCN) to the global boundary layer. In Amazonia, observations show that this downward transport can occur during strong convective activity. However, it is not clear from these studies over what spatial scale this cycle of aerosol formation and downward supply of CCN is occurring. Here, we aim to quantify the extent to

5 which the supply of aerosol to the Amazonian boundary layer is generated from nucleation within a 1000 km regional domain or from aerosol produced further afield, and the effectiveness of the transport by deep convection. We run the atmosphere-only configuration of the HadGEM3 climate model incorporating a 440 km × 1080 km regional domain over Amazonia with 4 km resolution. Simulations were performed over several diurnal cycles of convection. Below 1 km altitude in the regional domain, our results show that nucleation within the regional domain accounts for only 1.8 % of all Aitken and accumulation mode

aerosol particles, whereas nucleation that occurred outside the domain (in the global model) accounts for 81 %. The remaining aerosol is primary in origin. Above 10 km, the regional-domain nucleation accounts for up to 64 % of Aitken and accumulation mode aerosol, but over several days very few particles nucleated above 10 km in the regional domain are transported into the boundary layer within the domain, and in fact very little air is mixed that far down. Rather, particles transported downwards into the boundary layer originated from outside the regional domain and entered the domain at lower altitudes. Our model

results show that CCN entering the Amazonian boundary layer are transported downwards gradually over multiple convective cycles on scales much larger than 1000 km. Therefore, on a 1000 km scale in the model (approximately one-third the size of Amazonia), trace gas emission, new particle formation, transport and CCN production do not form a 'closed loop' regulated by the biosphere. Rather, on this scale, long-range transport of aerosol is a much more important factor controlling CCN in the boundary layer.



# 1 Introduction

Nucleation, or new particle formation (NPF), is important for aerosol-cloud interactions and thus climate, as the newly formed particles can grow to form cloud condensation nuclei (CCN) that affect cloud droplet number concentrations and cloud properties (Pierce and Adams, 2007; Merikanto et al., 2009; Wang and Penner, 2009; Kazil et al., 2010; Gordon et al., 2016; Dunne et al., 2016; Gordon et al., 2017). Global model studies have shown that NPF contributes around 54 % of the global present-day CCN in the boundary layer (Gordon et al., 2017), and that 35 % of the CCN were formed by NPF in the free and upper troposphere and later transported into the boundary layer (Merikanto et al., 2009). The downward transport can take place in large-scale subsidence in the general circulation such as in the Hadley cell, which is resolved by global models, or in the downdrafts of deep convection, which is a parameterised process in global models. These global-scale studies clearly show that high-altitude NPF contributes to low-level CCN, but the relative roles of these two transport mechanisms for the NPF-aerosol-CCN process is unknown. Here we aim to quantify the effectiveness of convective transport to influence aerosol particles in the boundary layer, inspired by measurements made over Amazonia showing a free-tropospheric source of aerosol into the boundary layer in a convective environment (Wang et al., 2016).

NPF involves inorganic species such as $H_2SO_4$-$H_2O$ and $H_2SO_4$-$NH_3$-$H_2O$ (Weber et al., 1995; Vehkamäki et al., 2002; Kirkby et al., 2011; Dunne et al., 2016) and the oxidation products of volatile organic carbon vapours such as monoterpenes producing HOMs, or highly oxygenated molecules, (Kirkby et al., 2016; Tröstl et al., 2016). Previous studies have found that NPF is affected by precursor gas concentrations as well as by temperature and the condensation sink. Low temperatures in the UT usually slow down the chemical reaction of extremely low volatility organic compounds (Simon et al., 2020), but also reduce the vapour pressure of the gas precursors and thereby enhance NPF (Zhao et al., 2020). Yu et al. (2017) also reported that the low-altitude CCN that are generated by NPF from $H_2SO_4$-$H_2O$ and organic gas molecules, were changed by 10 %-30 % when the temperature dependence of NPF was added. The condensation sink is also an important factor that affects the production and concentration of particles smaller than 3 nm in diameter (Kulmala et al., 2001a, b; Dal Maso et al., 2002) by modulating the concentration of nucleating and condensing vapours. Here we also explore the role of hydrometeors within deep convective clouds as a sink for the condensable gases (Kazil et al., 2011).

In Amazonia, it has been shown that NPF in the boundary layer was rarely observed and thus is insufficient to sustain CCN during the dry-to-wet transition season (Zhou et al., 2002; Krejci et al., 2003; Rissler et al., 2006; Spracklen et al., 2006; Rizzo et al., 2010; Andreae et al., 2018; Wimmer et al., 2018; Rizzo et al., 2018). However, aircraft measurements have shown that strong NPF in the upper troposphere (UT) (from precursor vapours transported upwards by deep convection) can create an abundant supply of small nuclei that, following downward transport and particle growth, could account for some boundary layer CCN (Clarke et al., 1998, 1999a, b; Clarke and Kapustin, 2002; Weigel et al., 2011; Wang et al., 2016; Andreae et al., 2018; Williamson et al., 2019). However, from the observations alone it is unknown whether the transport is predominantly via convective downdrafts on the spatial scale of the deep convective cells, or via large-scale subsidence that takes days to transport particles to lower altitudes.




The CCN concentration at 0.4 % supersaturation in the Amazonian boundary layer is around 185 cm$^{-3}$ in the wet season and
2500 cm$^{-3}$ during the dry season (Andreae, 2009). The total particle concentration is around 300 cm$^{-3}$ in the wet season and
3000 cm$^{-3}$ in the dry season (Andreae, 2009; Pöhlker et al., 2016; Pöhlker et al., 2018). Aircraft measurements show that more
than 80 % of the particles were between 20 and 90 nm in diameter in the UT during the dry season, suggesting that they are
formed by NPF (Andreae et al., 2018). Particle concentrations exceeding 20000 cm$^{-3}$ were observed above 8 km during the
ACRIDICON-CHUVA campaign in September and October 2014. Therefore, it is plausible that this high-altitude aerosol may
contribute to the aerosol populations in this region through downward transport in an environment with strong vertical motion.

Andreae et al. (2018) hypothesised that the newly formed aerosol particles in Amazonian UT could be mixed and transported
into the lower troposphere and contribute to boundary layer particles. Based on the observations from ACRIDICON-CHUVA
in the dry season, they proposed that the organic compounds in UT particles were derived from gas-phase oxidation of insol-
uble gas precursors which were emitted from the rainforest and transported upwards by deep convection. In the wet season,
based on GoAmazon2014/5 observations, Wang et al. (2016) concluded that the rapid vertical transport allowed newly formed
particles in the FT to enter the boundary layer in downdrafts associated with precipitation.

To understand the formation of aerosol and its vertical transport in a convective environment, it is necessary to use a model
that resolves cloud motion. Global models have shown that the UT is a major source of CCN in the boundary layer (Merikanto
et al., 2009), but the results may only be reliable in regions where aerosol is transported in the types of synoptic-scale cir-
culation that are resolved by the model, such as in the descending branch of the Hadley cell in sub-tropical regions. Deep
convection is a regional- to local-scale system. Due to the strong vertical velocities, deep convection can potentially transport
particles and vapours upwards and downwards on scales of a few kilometers that are unresolved by a global model (global
models parameterize the vertical exchange of trace gases, but they do not resolve coherent updrafts and downdrafts and the
associated clouds). Most studies have focused on upward transport. Ekman et al. (2004) used a cloud-resolving model and
found that only small particles (5.84 to 31.0 nm) were transported to the UT by a deep convective updraft, while larger par-
ticles were scavenged. Some of the smaller particles eventually grew and served as CCN or ice-nucleating particles (INP).
Using an axisymmetric dynamic cloud model Yin et al. (2005) reported that aerosols were transported from the boundary layer
to mid-cloud level and contributed to the aerosol mass within the hydrometeors in the deep convection. The analysis of Zhao
et al. (2020) of ACRIDICON-CHUVA and GoAmazon2014/5 observations using a regional-scale chemical transport model
with a detailed treatment of organic vapours suggested the importance of fast convective transport for low-altitude particles and
CCN. However, to understand the spatial scales over which the NPF-aerosol-CCN pathway takes place, i.e. over a large-scale
or within a 1000 km regional convective domain, we need to combine a global and regional model to represent both processes.

Inspired by the studies outlined above, we address the following questions:

(1) How do NPF-deep convection interactions on regional scales affect the vertical distributions of particles in Amazonian
tropical rainforest?



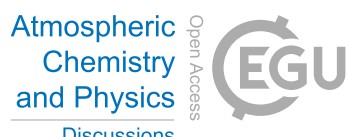

(2) How much does NPF occurring on a regional scale (∼1000 km) affect the CCN budget of the region? What fraction of CCN is created regionally versus being transported into a region from outside?

(3) How effective are deep convective clouds at transporting particles downwards to low-levels, and on what spatial and temporal scales is the process occurring?

(4) How are particles, nucleation and growth rates in the regional domain sensitive to changes in nucleation mechanisms?

The previous studies are well designed and help to understand aerosol sources on global and regional scales. However, to address our questions about the relative importance of regional and global-scale transport we use a model that combines high resolution (to resolve convection) with interactive aerosol microphysics that represents aerosol diameters as low as 3 nm and allows the simulation of NPF and growth. This is accompanied by a broader view from the global model. We simulate 3 days of the ACRIDICON-CHUVA campaign as a case study and investigate the role of deep convection in supporting the UT NPF and the boundary layer aerosols.

The sections are organised as follows. Section 2 introduces the observations and models used in our study. The results are analysed in Sect. 3 and the first subsection 3.1 shows the comparisons with the observations. Section 3.2 has an overview of aerosols, gas precursors, nucleation and growth rates in Amazonia, and it briefly explores the sensitivity of them to different nucleation mechanisms, oxidation rates and emission rates. Section 3.3 describes the effects on NPF of adding a contribution to the condensation sink from cloud droplets and ice crystals. Section 3.4 quantifies the contributions of NPF from the global and regional models to aerosols in the 1000 km by 440 km regional domain. Section 3.5 further explores the vertical transport by deep convection in the regional model. We finally discuss and conclude the results in Sect. 4.

## 2 Methods

### 2.1 Observations

#### 2.1.1 ACRIDICON-CHUVA campaign

Our study is motivated by the measurements made during ACRIDICON-CHUVA (Aerosol, Cloud, Precipitation, and Radiation Interactions and Dynamics of Convective Cloud Systems-Cloud Processes of the Main Precipitation Systems in Brazil: A Contribution to Cloud Resolving Modeling). We also use measurements from the GPM (Global Precipitation Measurement) satellite mission, and monoterpenes observations from aircraft measurements and ATTO (Amazon Tall Tower Observatory) tower (Kuhn et al., 2007; Yáñez-Serrano et al., 2018; Zannoni et al., 2020). The aim of ACRIDICON-CHUVA was to study the relationships between trace gases, particles and radiation in Amazonian convective environment. The campaign included 14 flights from early September until the beginning of October in 2014, centred around Manaus in Brazil (3.1° S, 60.0° W). They measured cloud, aerosol and trace gas properties in forest, urban and marine environments using the HALO aircraft (Wendisch et al., 2016; Andreae et al., 2018); see Fig. 1.





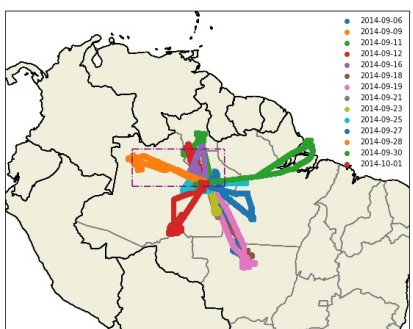

**Figure 1.** A map of the flight tracks during ACRIDICON-CHUVA. The flight track in purple is AC11 (on 16 September 2014), and the purple dashed box denotes the regional domain used in this study.

### 2.1.2 Meteorology

The campaign took place during the transition season in Amazonia, which was towards the end of the dry season and the onset of the wet season. General subsidence caused by a northward-shifted ITCZ dominated the dry season during this period, but during the transition in September, moisture advection from the Atlantic Ocean was sufficient to cause the large-scale

circulation to shift leading to an increase in rainfall (Li and Fu, 2004). North-easterly and easterly wind dominated this period, bringing in moist air from the South Atlantic Ocean (Martin et al., 2016). The surface temperature was at its highest for the year in September 2014 and moisture started to increase from September onwards. The monthly averages of surface temperature reached 28 °C and the specific humidity was over 19 g kg$^{-1}$ (Collow et al., 2016). The UT equivalent potential temperature in September ranged from 60 °C to 80 °C and relative humidity was around 20 % in September and rose to 100 % in early

October favouring the development of deep convection (Collow et al., 2016). Warm sea surface temperatures occurred during the campaign (Martin et al., 2016; Andreae et al., 2018).

During our simulation period (16 to 18 September 2014), MODIS images show that on 16 September, the sky was partially cloudy with shallow cumulus clouds. A deep convective cell formed to the northeast of Manaus with a large anvil above 12 km in altitude on 17 September. On this day, cloud fraction at the location of our regional domain reached 100 %. A squall line

passed Manaus and it extended as far as 240 km to the northwest of Manaus. On 18 September 2014, the sky became partially cloudy and the convective cells were diminished or left the region.

### 2.1.3 Aerosol measurements

The measurements of particle number concentrations from ACRIDICON-CHUVA are used in this study. The instruments aboard the HALO aircraft measured aerosol particle concentrations up to around 14 km in altitude, approximately where

boundary of the upper troposphere and lower stratosphere locates, and the flight area covered the region of interest (Fig.





2) as well as wider regions of Amazonian basin. Four butanol-based CPCs (condensation particle counters) and a UHSAS (Ultra-High Sensitivity Aerosol Spectrometer) were used to measure particles of various sizes. The measured particles are split into two size ranges by diameter: those larger than 20 nm dry diameter ($N_{D>20nm}$) and those larger than 90 nm diameter ($N_{D>90nm}$). The $N_{D>20nm}$ data consists of measurements with lower cut-off diameters that vary with pressure because of inlet

loss: 9.2 nm at 1000 hPa, 11.2 nm at 500 hPa, and 18.5 nm at 150 hPa (Andreae et al., 2018). $N_{D>90nm}$ were measured using a UHSAS and an OPC (optical particle counter). For a more extensive description of the measurements, see Andreae et al. (2018).

## 2.2    Models and simulations

### 2.2.1    Global and regional model configurations

We use the Hadley Centre Global Environment Model version 3 (HadGEM3), which is based on the Unified Model (UM) framework, and incorporates the United Kingdom Chemistry and Aerosol (UKCA) model. The model includes a regional nest of configurable spatial resolution that is one-way coupled to the global model, which allows the global model to affect the processes in the regional domain but the regional domain does not affect the global model. The UKCA model was first run at kilometer-scale resolution by Planche et al. (2017) and tested in the one-way nesting configuration by Gordon et al. (2018).

The global model is based on GA7.1 (Global Atmosphere v7.1) of UM11.3 with the Even Newer Dynamics for General atmospheric modelling of the environment (ENDGame) dynamics (Wood et al., 2014). In the global model we use the N216 grid ($\sim$ 65 km horizontal resolution) with 70 vertical levels up to 80 km altitude. Sea surface temperatures are fixed and the land surface is represented by the JULES model (Walters et al., 2019).

The nested regional domain is centred at (1.5°S, 63°W) and has 4 km horizontal resolution. The domain size is 440 km north-

south and 1080 km east-west (Fig. 2) to align approximately with the mean wind direction. There are 70 vertical levels to 40 km altitude, with 63 levels in the lowest 20 km, the region of interest of this study. The residence time of air in the regional domain is determined by horizontal wind speed and is between 20 and 40 hours, which is around half of the total simulation time. The regional model is driven by hourly boundary conditions generated from the global model that provides meteorology (temperature, 3D wind velocity, cloud liquid and ice water, humidity and rain) as well as aerosol and trace gas boundary

conditions for the regional model.

Convection is parameterised in the global model and is resolved in the regional model. The resolved convection allows explicit heat transfer and tracer transport, whereas the parameterisation simplifies the transport processes (Fritsch and Chappell, 1980; Gregory and Rowntree, 1990; Stratton et al., 2009; Derbyshire et al., 2011; Walters et al., 2019).

UKCA uses the GLOMAP-mode aerosol scheme which produces, grows, transports, and removes aerosol (Mann et al., 2010,

2014). The two-moment aerosol microphysics scheme has five log-normal modes of variable number and size (but fixed width) to define aerosol size distributions. Four of these are soluble modes comprising nucleation, Aitken, accumulation and coarse





size modes. The fifth mode is an insoluble Aitken mode. The simulated aerosol is made up of sulfate, sea-salt, black carbon and organic carbon. UKCA includes organic carbon and black carbon emissions from biomass and fossil fuel burning from the GFED version 3.1 and CMIP5 inventories (Van Der Werf et al., 2003; Kanakidou et al., 2005). UKCA uses monthly averages

of $SO_2$ and DMS from CMIP5 emission inventories in both the global and regional models. The marine source of DMS has been parameterised based on Kettle et al. (1999) and land biomass burning (van der Werf et al., 2006; Lamarque et al., 2010; Granier et al., 2011; Diehl et al., 2012); $SO_2$ comes from volcano eruptions (Andres and Kasgnoc, 1998; Halmer et al., 2002), biomass burning (GFEDv3.1 inventory), bio-fuel burning, fossil-fuel burning and industrial emissions (Cofala et al., 2005). The monoterpene emissions mainly come from the monthly averages of vegetation (Guenther et al., 1995).

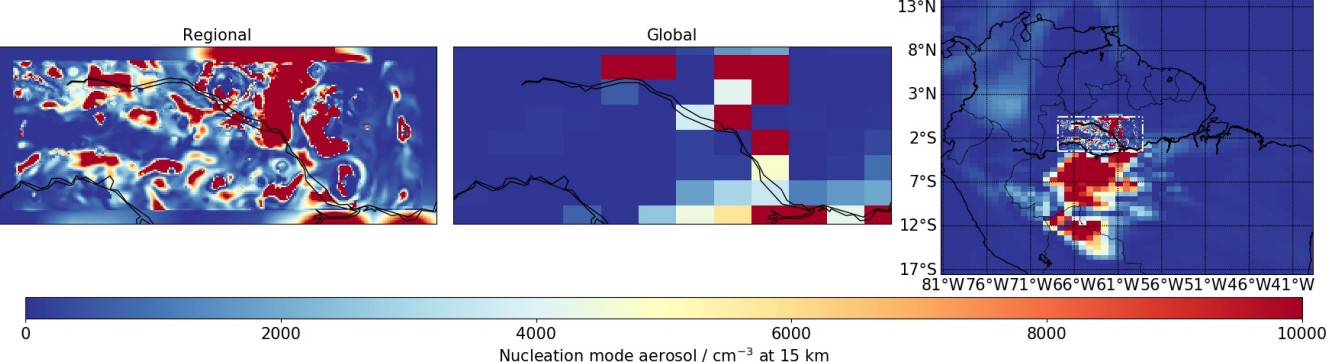

**Figure 2.** Maps of nucleation mode aerosol number concentrations from the regional (left) and global models (middle) at a height of 15 km and at 15 UTC on 17 September, 2014. The map on the right presents a broader view of South America and shows the location of the regional domain (dotted dashed box).

The global model uses the single-moment cloud microphysics scheme of Wilson and Ballard (1999). It activates aerosol particles following Abdul-Razzak and Ghan (2000) which considers the probability distribution function (PDF) of updraft velocities centred around the large-scale vertical velocity to derive cloud droplet number concentrations for each time step (West et al., 2014). The regional model uses the Cloud-AeroSol Interacting Microphysics (CASIM) model. CASIM is a two-moment cloud microphysics scheme that includes cloud droplets, rain, ice, snow and graupel (Shipway and Hill, 2012; Hill et al., 2015;

Grosvenor et al., 2017). All hydrometeor distributions are defined by gamma distributions. CASIM activates aerosol particles to form cloud droplets depending on the mean updraft velocity in the gridbox (Grosvenor et al., 2017; Miltenberger et al., 2018). The droplet number concentrations are prognostic, which means that they are stored at each timestep, but if the model activates more droplets in the new timestep than the previous one, the old number concentration is overwritten (Gordon et al., 2020).

The UKCA and CASIM models are coupled to allow UKCA to pass aerosol particle number and mass to CASIM for activation, with the mass of different chemical components (sea-salt, sulfate, organic carbon and black carbon) used to derive the hygroscopicity for aerosol activation in CASIM (Gordon et al., 2020). CASIM also passes the rates of autoconversion and



accretion to UKCA in order to affect the convective scavenging of aerosols by precipitation (Miltenberger et al., 2018).

Aerosol particles are removed by rain formation via collision coalescence of droplets, as well as by rain impaction scavenging.
Precipitation removes particles of various sizes determined by a collection efficiency look-up table (Mann et al., 2010; Kipling et al., 2013). In the original version of the model, these processes scavenge only those particles larger than 10 nm in diameter. Here we also investigate the effect of cloud hydrometeors on NPF through their effect on the condensation sink, as described in Sect. 2.2.2.

### 2.2.2   New particle formation

In the UKCA model, NPF produces new particles at 3 nm in diameter. The schematic diagrams in Fig. 3 show the oxidation, nucleation and particle growth pathways of the binary and biogenic nucleation mechanisms which we use in this study.

Binary sulfuric acid-water and pure biogenic nucleation mechanisms are mainly used in this study. Binary nucleation follows the parameterisation of sulfuric acid-water ($H_2SO_4$-$H_2O$) in Vehkamäki et al. (2002). In the UKCA model, the binary nucleation precursor gas $H_2SO_4$ mainly comes from the oxidation of $SO_2$ and DMS. It also condenses on to existing aerosols, contributing to their growth.

We also added the biogenic nucleation mechanism along with its precursor gas (HOMs) from the parameterisation in Gordon et al. (2016), which was based on the CLOUD chamber experiments (Kirkby et al., 2016). The experiments found and quantified pure biogenic nucleation, which required only HOMs to form new particles (Kirkby et al., 2016), and subsequent growth with oxidised BVOCs and $H_2SO_4$-$H_2O$. This mechanism has been applied to several models (Gordon et al., 2016; Zhu et al., 205  2019; Zhao et al., 2020). Here, we use it to simulate Amazonian NPF, focussing on the environmental conditions for, and the consequences of, new particle formation rather than the chemical mechanism.

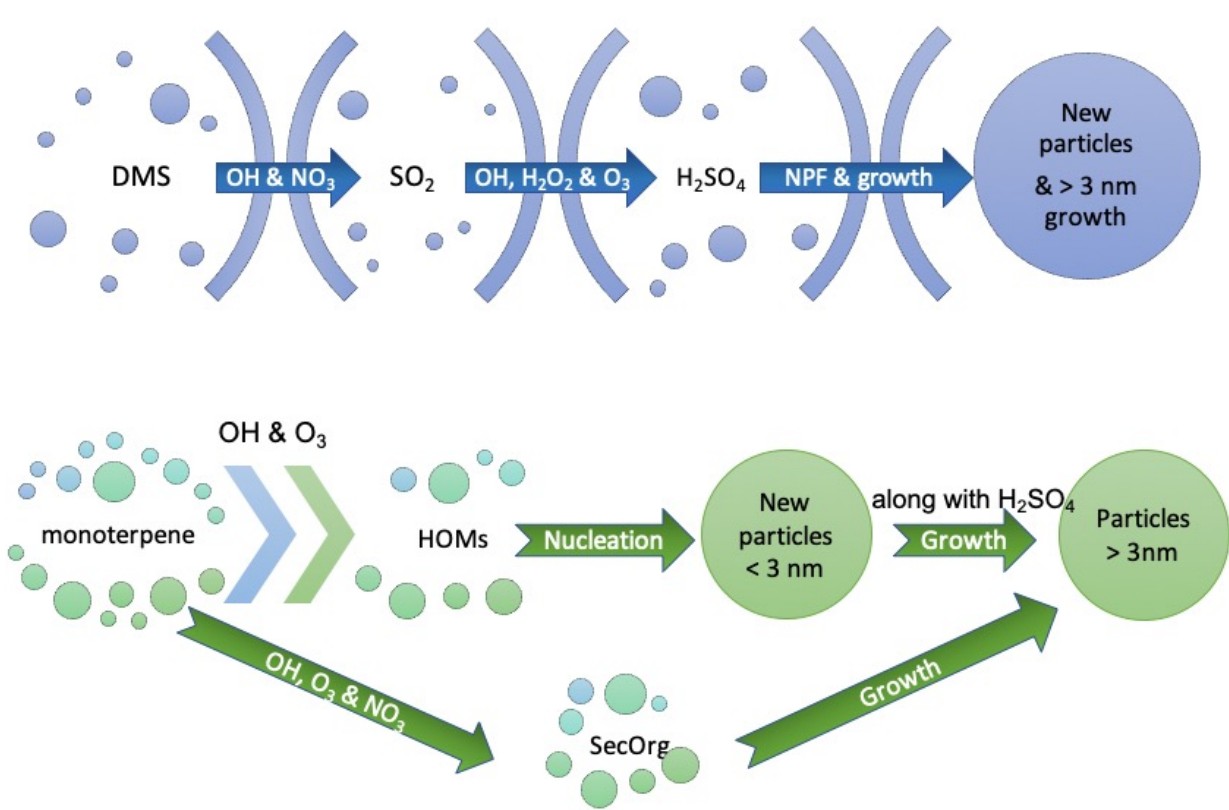

**Figure 3.** Schematic diagrams showing oxidation pathways and their roles in NPF and particle growth in the UKCA for binary nucleation (upper panel) and biogenic nucleation (lower panel).

In the original UKCA setup, monoterpenes are oxidised to SecOrg (secondary organics) which can grow particles by condensation. With the new parameterisation, monoterpenes are first oxidised to SecOrg, then the rest of the monoterpenes in the environment is used to derive HOM concentrations using the same steady-state approximation as used by Gordon et al. (2016). A simplified (offline) chemistry scheme is used to reduce computational cost. Here, the oxidants (OH, $O_3$, $HO_2$, $H_2O_2$ and $NO_3$) are read in from monthly mean ancillary files generated from a full chemistry simulation (Walters et al., 2017), then OH and $HO_2$ concentrations are modulated according to the diurnal cycle of solar radiation. The chemical reactions and rate constants are summarised in Table 1. In the Amazonian environment, isoprene is another BVOC that significantly affects upper tropospheric aerosol mass (Schulz et al., 2018), but the simplified chemistry scheme does not include isoprene and the related chemistry. This representation of the chemical mechanism is one of many uncertainties associated with biogenic particle formation. Our definition of HOMs is based on the chamber experiment from CLOUD (Ehn et al., 2014; Kirkby et al., 2016;





Bianchi et al., 2019) which found that a wide range of highly oxidised organic vapours were able to nucleate particles and that particle growth varies depending upon the volatility of the organic vapours (Tröstl et al., 2016; Stolzenburg et al., 2018). The experiments showed that HOMs can be formed from alpha-pinene, which is a subset of the monoterpenes, by reaction with
OH and $O_3$. The yields of HOMs from monoterpenes by OH and $O_3$ are is 1.2 and 2.9 %, respectively (Gordon et al., 2016).

**Table 1.** The chemical reactions in the offline chemistry and the corresponding rate constants.

| Reactions | Rate constants |
|---|---|
| DMS + OH -> $SO_2$ | $9.6 \times 10^{-12}$ |
| DMS + OH -> $SO_2$ + DMSO | $3.04 \times 10^{-12}$ |
| DMS + $NO_3$ -> $SO_2$ | $1.9 \times 10^{-13}$ |
| DMSO + OH -> $SO_2$ | $5.8 \times 10^{-11}$ |
| $SO_2$ + OH -> $H_2SO_4$ | $3.00 \times 10^{-31}$ |
| Monoterpene + OH -> SecOrg | $1.2 \times 10^{-11}$ |
| Monoterpene + $O_3$ -> SecOrg | $1.01 \times 10^{-15}$ |
| Monoterpene + $NO_3$ -> SecOrg | $1.19 \times 10^{-12}$ |
| $HO_2$ + $HO_2$ -> $H_2O_2$ | $2.2 \times 10^{-13}$ |
| OH + $H_2O_2$ -> $H_2O$ | $2.9 \times 10^{-12}$ |

In the UKCA model, HOM concentrations are used to derive the nucleation rate of particles below 1.7 nm in diameter. The pure biogenic nucleation rate at 1.7 nm in our model follows Gordon et al. (2016). It is the sum of neutral and ion-induced nucleation rates and is given by:

$$J_{1.7nm} = \exp(-(T-278)/10) \times (A_1 \times ([HOM]/10^7)^{A_2 + \frac{A_5}{[HOM]/10^7}} + 400 \times A_3 \times ([HOM]/10^7)^{A_4 + \frac{A_5}{[HOM]/10^7}}), \qquad (1)$$

where $J_{1.7nm}$ is the nucleation rate in $cm^{-3}$ $s^{-1}$, HOM is the concentration of the pure biogenic nucleation gas precursor in molecules per $cm^{-3}$, and $A1-5$ are constant parameters. Both neutral and charged rates are multiplied by $\exp(-(T-278)/10)$ to model temperature dependence; we note, however, that this dependence is uncertain and was initially proposed only as a sensitivity study by Gordon et al. (2016). We also assume here a constant value of $400\,cm^{-3}$ for the ion concentration. Particles of 1.7 nm diameter grow to 3 nm, through the condensation of HOM and $H_2SO_4$-$H_2O$ based on Kerminen and Kulmala (2002).

**2.2.3   Simulation details**

We run both the global and regional models from 16 to 18 September 2014, which is close to the end of Amazonia dry season and is the time when the ACRIDICON-CHUVA field campaign took place (Wendisch et al., 2016). The global model was spun-up for 15 days (1-15 September 2014) in order to allow the model to initialise the aerosol fields.

During the 3-day simulation, deep convection usually occurs at 15 UTC (11 LT) and reaches a maximum two hours later. The
domain averaged surface rain rate reaches a maximum (equivalent to around 118 mm $hr^{-1}$) within an hour after the start of





the deep convection. The rain lasts for 5 to 6 hours, then the convective clouds start to dissipate and completely disappear by midnight. During the most vigorous phase, cloud top height reaches a maximum of 20 km in altitude. In the initial stages of cloud development, at below 2 km in altitude, the cloud coverage is around 50 %-70 %. As the clouds deepen, the low-level clouds are transformed into deep clouds within an hour, with the low-level cloud cover being reduced to approximately 10 %.

At the same time, mid-level cloud covers around 10 % of the horizontal domain and the high-level cloud fraction reaches 100 %.

In this paper, we test the binary and pure biogenic nucleation mechanisms and investigate the sensitivity of the particle number concentrations to the nucleation rate, oxidation rate, emission rate and condensation sink. We also study the source and the vertical distribution of the particles in the regional domain. Table 2 shows the name and components of the nucleation mecha-

nisms used for all the simulations in this study. The first five simulations use (1) binary sulfuric acid-water nucleation (denoted as Bn), (2) binary nucleation with the nucleation rate increased by a factor of 10 (Bn×10), (3) pure biogenic nucleation from Gordon et al. (2016) (Bio), (4) pure biogenic nucleation with the oxidation rates of monoterpenes reduced by a factor of 10 (BioOx), and (5) reduced monoterpenes oxidation and the monoterpenes emission rate increased by a factor of 10 (BioOxEm).

**Table 2.** The table of all simulations and the NPF mechanisms. monoterpenes (MT) oxidation÷10 denotes that the oxidation rates of monoterpenes (to secondary organics) are reduced by a factor of 10, and 10×MT emission denotes increasing the monoterpenes emission rate by a factor of 10. CCS = condensation sink from clouds.

|  | Binary | Biogenic | MT oxidation÷10 | 10×MT emission | CCS | NPF notes |
|---|---|---|---|---|---|---|
| 1. Bn | ✓ |  |  |  |  |  |
| 2. Bn×10 | ✓ |  |  |  |  | Bn nuc rate×10 |
| 3. Bio |  | ✓ |  |  |  |  |
| 4. BioOx |  | ✓ | ✓ |  |  |  |
| 5. BioOxEm |  | ✓ | ✓ | ✓ |  |  |
| 6. BioOxCCS |  | ✓ | ✓ |  | ✓ |  |
| 7. BioOxEmCCS |  | ✓ | ✓ | ✓ | ✓ |  |
| 8. off_allNPF |  | ✓ | ✓ | ✓ | ✓ | NPF off everywhere |
| 9. off_regNPF |  | ✓ | ✓ | ✓ | ✓ | NPF off in regional |
| 10. NPF_1-4km |  | ✓ | ✓ | ✓ | ✓ | NPF 1-4 km only |
| 11. NPF_4-7km |  | ✓ | ✓ | ✓ | ✓ | NPF 4-7 km only |
| 12. NPF_7-10km |  | ✓ | ✓ | ✓ | ✓ | NPF 7-10 km only |
| 13. NPF_10-13km |  | ✓ | ✓ | ✓ | ✓ | NPF 10-13 km only |
| 14. NPF_13-16km |  | ✓ | ✓ | ✓ | ✓ | NPF 13-16 km only |

The justification for the changes in monoterpenes emissions is that biogenic volatile organic compounds (BVOCs) usually

have various species and a wide range of abundances and volatilities, and the rates of BVOC emissions and their oxidation





mechanisms are still not well understood despite some progress reported in the literature Sindelarova et al. (2014). Additionally, the oxidation rate of monoterpenes itself has large range of uncertainty (up to $10^{-10}$) and the rate even differs by three orders of magnitude with the same oxidant (Kwok and Atkinson, 1995). Our comparisons with observed aerosol (Sect. 3.1) suggest that monoterpenes are oxidised too quickly in the default simulation (Bio) and are not transported to the UT where they

could contribute to NPF. Therefore, we reduce the oxidation rates in the UKCA to allow for a longer monoterpenes lifetime so that monoterpenes will be more likely to contribute to NPF (BioOx simulation). We do not increase the oxidation rates because they will drive the simulations away from the observations by producing too few aerosols in the UT.

The averaged monoterpenes mixing ratios in the regional domain overestimate the ATTO tower observations at the surface by a factor of 2.25, by a factor of 2.73 at 75 m and by a factor of 2 at 155 m (Yáñez-Serrano et al., 2018; Zannoni et al., 2020). In

contrast, between 1 and 2.5 km, the simulations underestimate aircraft measurements by a factor of 0.77 (Kuhn et al., 2007). There are no measurements of monoterpenes available in the UT, but we expect even stronger underestimation there. The insufficient mixing ratios of monoterpenes are likely the cause of the very low nucleation rates ($1.8 \times 10^{-3}$ cm$^{-3}$ s$^{-1}$) in the UT in our simulations with the default biogenic nucleation scheme and with reduced oxidation rates, such that these simulations cannot reproduce the observed concentrations (Sect. 3.1 and 3.2). For example, our BioOxCCS simulation underestimates the

observed particle number concentrations at 12 km by a factor of 8.6. Therefore, for the simulation with reduced oxidation rates, we also increase the monoterpenes emission rate by a factor of 10 to allow more monoterpenes to be transported to the UT to further enhance NPF (BioOxEm simulation). This is our default simulation for most of the rest of the study to explore the factors controlling NPF and aerosol transport.

In September 2014, strong biomass burning events took place, which led to high condensation sinks and partly explained the

reason why no NPF events were observed close to the surface (Andreae et al., 2018). However, our models do not capture the suppression due to a lack of high-resolution biomass burning emissions and the overestimated monoterpenes emission at surface. Thus, we eliminate NPF below 100 m for all simulations here.

The second set of simulations is designed to examine the effect on NPF of the condensation sink on cloud particles. The default model includes only a sink of vapours and nuclei onto existing aerosol but not onto cloud hydrometeors. We implement an

additional condensation sink from cloud droplets and ice crystals, and add the value to the condensation sink from existing aerosols, which then affects the nucleation rate (Kazil et al., 2011). The cloud hydrometeor condensation sink is defined by assuming a fixed number concentration (100 cm$^{-3}$) of cloud droplets and ice particles to calculate radii that are fed into the Fuchs and Sutugin (1971) expression $CCS1 = N_d(r_{cloud} + r_{ice})$, where CCS1 is the condensation sink with units of m$^{-2}$, $N_d$ is the number concentration of droplets and ice, and $r_{cloud}$ and $r_{ice}$ are the radii. The equivalent sink with units of s$^{-1}$ is obtained

by multiplying by the gas diffusion coefficient $CCS = 4\pi D_v \times CCS1$, where $D_v$ is the gas-phase diffusion coefficient of the vapour.

To understand the source of particles in the regional domain, we ran additional simulations in which nucleation was switched





off in both the regional and global models (off_allNPF) and in the regional model only (off_regNPF). These simulations allow us to quantify the effect of NPF within the 1080 km by 440 km regional model domain compared to that from outside of the

regional domain.

A final set of five simulations was performed to understand how particles that are nucleated at a particular altitude are transported vertically and thereby affect aerosol at other altitudes. These simulations are also based upon the BioOxEmCCS simulation. For these five simulations, we allow NPF at all heights above 100 m in the global model, but only allow NPF in the regional model to occur at certain altitudes (1-4 km, 4-7 km, 7-10 km, 10-13 km and 13-16 km).

## 3 Results

### 3.1 Model-observation comparison

Figure 4 shows the measured profiles of median particle number concentrations from flight AC11 (16 September 2014) of the ACRIDICON-CHUVA campaign compared to five of the model simulations. All the data in this section are converted to standard temperature (273.15 K) and pressure ($10^5$ Pa; STP), using Eq. (2).

$$N_{STP} = \frac{N_{ambient} \times 10^5 \times T}{p \times 273.15},$$
(2)

where $N_{STP}$ is the number concentrations of particles converted to STP, $N_{ambient}$ is the number concentration at the current temperature and pressure, T is temperature in K, and p is the pressure in Pa. We use the data from flight AC11 because the flight track falls well within the regional domain and the date of measurement is within the simulation period. Supplementary Fig. A1 shows the full campaign.

Below 3 km in altitude, the observed median $N_{D>20nm}$ is homogeneous with height, with a concentration of around 1600 cm$^{-3}$. The median of $N_{D>20nm}$ then increases with altitude to a maximum at 11.8 km with a value of 19000 cm$^{-3}$. The observed profile of median $N_{D>90nm}$ is also homogeneous with height up until around 2 km and has a similar median concentration to $N_{D>20nm}$ (1400 cm$^{-3}$), which shows that the observed concentration of small particles is low at these low altitudes. Above 2 km, the observed $N_{D>90nm}$ decreases to around 6-8 km and then increases again with height and reaches around 1100

cm$^{-3}$ at 12.5 km.





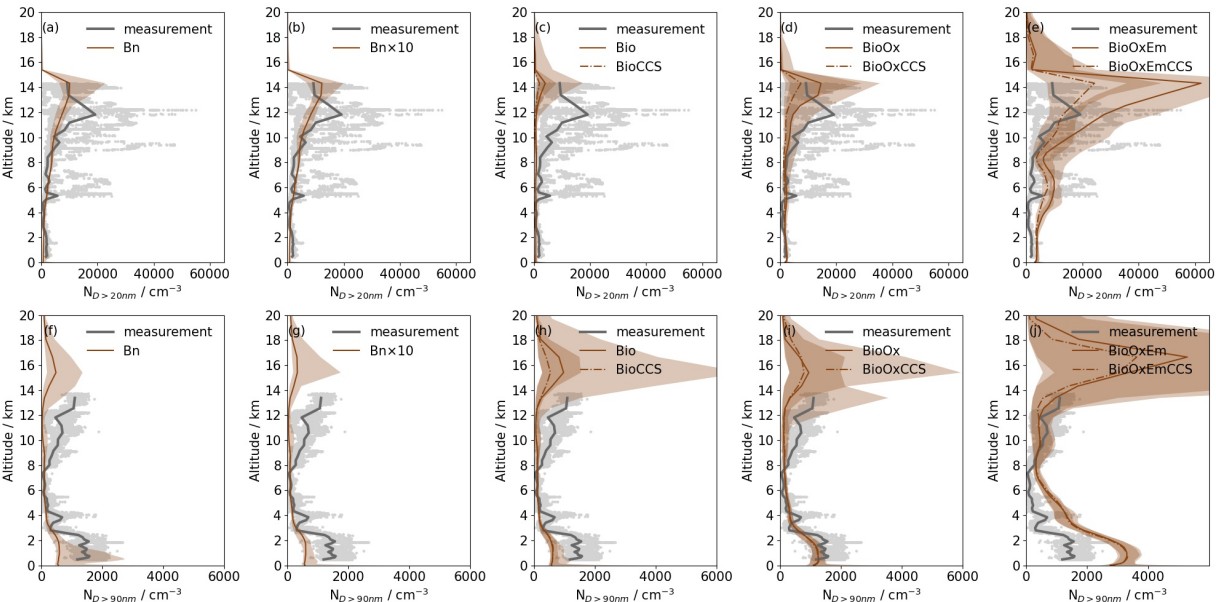

**Figure 4.** The observed and modelled vertical profiles of median number concentrations of particles with diameters >20 nm ($N_{D>20nm}$, top row) and >90 nm ($N_{D>90nm}$, bottom row). The observations are shown as grey dots and a grey line (repeated for all panels). The grey dots are individual observations from ACRIDICON-CHUVA flight AC11 (16 September 2014) with a time resolution of 1 minute, and the thick grey lines are the medians of the observations binned within the same height ranges as the regional model levels. The modelled results are from the various regional simulations averaged from 0 UTC on 17 September to 23 UTC on 18 September 2014, (a) Bn, (b) Bn×10, (c) Bio and BioCCS (dashed line), (d) BioOx and BioOxCCS (dashed line), and (e) BioOxEm and BioOxEmCCS (dashed line). The shading represents 2.5 % and 97.5 % percentiles from the modelling results.

In the simulation with the default binary nucleation mechanism (simulation Bn) and in that with a 10 times enhanced nucleation rate (Bn×10; Fig. 4 a and b) median number concentrations of $N_{D>20nm}$ are low at the surface, and then increase with height until they reach a maximum (9900 cm$^{-3}$) at 14 km in altitude where they start to decrease to almost zero at around 15 km in altitude. The two simulations exhibit similar number concentrations until 12 km. Throughout most of the profiles, both simulations reproduce the measurements well with an overall mean difference of -4.6 % (Bn) and 2 % (Bn×10). However, between 10 and 13 km the models underestimate the observations by 46 % for the Bn and and 37 % for the Bn×10 simulations. The profiles of $N_{D>90nm}$ (Fig. 4 f and g) show the highest concentrations below 2 km where the values are approximately constant with height (600 cm$^{-3}$). There is another peak at around 15.5-16 km (∼500 cm$^{-3}$) in altitude, and the concentrations are much lower between 4 and 13 km (< 100 cm$^{-3}$). These two simulations underestimate the observed $N_{D>90nm}$ by 46 % (Bn) and 47 % (Bn×10) when averaged over all altitudes.

In the simulations with biogenic nucleation (Bio, BioOx and BioOxEm) the median $N_{D>20nm}$ have low concentrations from the surface to around 10 km in altitude, where the particle number concentrations significantly increase. The profiles of $N_{D>90nm}$ have higher concentrations in the boundary layer and UT than at the altitudes in between. The Bio and BioCCS simulations



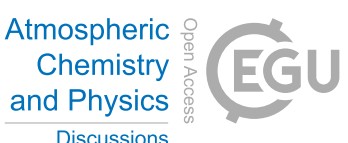

(Fig. 4 c) underestimate the height-averaged observed $N_{D>20nm}$ (by 52 %) and $N_{D>90nm}$ (by 36.7 %) suggesting insufficient NPF and particle growth. When we reduce the monoterpenes oxidation rates (BioOx and BioOxCCS; Fig. 4 d), the aerosol number concentrations increase because the reduced oxidation rate enables the longer-lived monoterpenes to be transported to the UT. The BioOx and BioOxCCS simulations therefore match the observed concentrations below 8 km, but still underestimate concentrations above 8 km by an average of 28 % (BioOx) and 50 % (BioOxCCS) for $N_{D>20nm}$ and 76 % for $N_{D>90nm}$ in both

simulations. With an increased monoterpenes emission rate (BioOxEm and BioOxEmCCS simulations; Fig. 4 e) the model produces significantly higher particle concentrations than in the other simulations, as expected. The BioOxEm simulation overestimates $N_{D>20nm}$ at all altitudes with an averaged overestimation of a factor of 3 for heights below 14.3 km, and overestimates $N_{D>90nm}$ below 9 km by an average factor of 3 (Fig. 4 j). Adding the cloud condensation sink (BioOxEmCCS) improves overestimation of $N_{D>20nm}$ above 9 km and the modelled concentration is reduced to around 30 % compared to the

observations. The increased emission rate combined with the cloud condensation sink allows the model to reproduce the UT aerosol number concentrations but causes too many particles in the lower troposphere. Whether these particles are formed by NPF within the regional model or in the global model is discussed in Sect. 3.4.

The simulations with binary nucleation mechanisms (Bn and Bn×10) produce $N_{D>20nm}$ and $N_{D>90nm}$ in the UT that are up to 100 times smaller than the three simulations with biogenic nucleation (Bio, BioOx and BioOxEm). The smaller concentrations

and variability occur because binary nucleation is determined by the $SO_2$ gas field that, due to its long lifetime relative to monoterpenes, is more controlled by the global model, whereas biogenic nucleation is controlled more by convective transport, mixing and oxidation in the regional model. We also ran a simulation in which the regional $SO_2$ emission was removed and the $SO_2$ profiles were almost identical to the Bn simulation meaning that the $SO_2$ in the regional domain was hardly affected by the regional-scale processes.

The BioOxEmCCS simulation is chosen as the base model for the rest of this study for two reasons: (1) it matches the observed particle concentrations well in the UT (Sect. 3.1); (2) it includes the suppression of unrealistic NPF inside clouds via the cloud condensation sink. Various factors including oxidation rates, oxidant concentrations, emissions and the condensation sink affect the model performance and these simulations can only give us a limited view of this sensitive environment since they lack the complexity to represent all of the processes that happen in reality. However, the overall reasonable match to observations shows

that the chosen model is well suited to addressing the aims of this study.

## 3.2 Analysis of particle formation and growth

Figure 5 shows vertical profiles of particle concentrations, nucleation and growth rates, and trace gas volume mixing ratios. All the profiles are averaged from 0 UTC on 17 September to 23 UTC on 18 September 2014. For the rest of the paper, ambient particle concentrations are quoted without the conversion to STP as performed for Fig. 4.






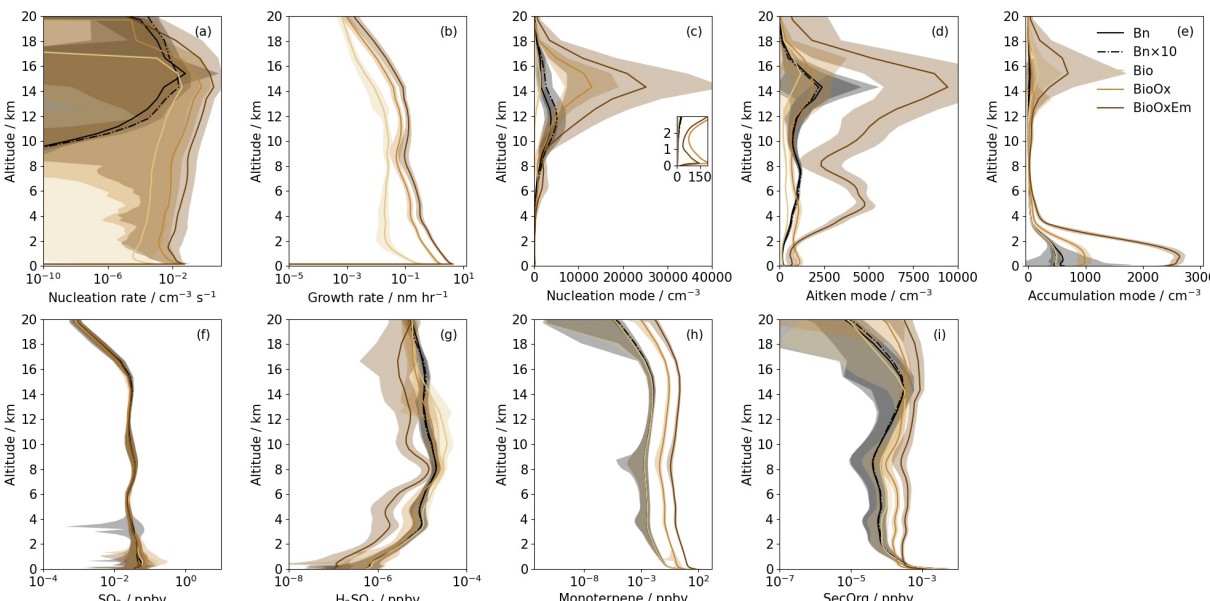

**Figure 5.** Regional domain- and time-averaged vertical profiles of ambient (a) nucleation rate (up to 3 nm in diameter), (b) growth rate in the biogenic nucleation mechanism (from 1.7 nm to 3 nm), (c) nucleation mode aerosol number concentrations (with the inset figure showing the number at lowest 3 km), (d) Aitken mode aerosol number concentrations, (e) accumulation mode aerosol number concentrations, (f) $SO_2$ volume mixing ratios, (g) $H_2SO_4$ volume mixing ratios, (h) monoterpenes volume mixing ratios, and (i) secondary organic (SecOrg; the oxidation product of monoterpene) volume mixing ratios. The results are from the simulations with binary nucleation (Bn; black solid), binary nucleation with 10 times nucleation rate (Bn×10; black dotted dashed), pure biogenic nucleation (Bio; light brown), biogenic nucleation with reduced (÷10) oxidation rate (BioOx; brown), and biogenic nucleation with reduced oxidation rate and enhanced (×10) monoterpenes emission (BioOxEm; dark brown). The shading represents one standard deviation either side of the mean at each height.

The NPF rates at 3 nm in diameter in all five simulations increase with height until 14.3 km, reaching a maximum of 3.5 cm$^{-3}$ s$^{-1}$ in the simulation with the most intensive nucleation (BioOxEm). The binary nucleation rates increase more sharply with height because of the strong temperature dependence of the binary nucleation rate (Vehkamäki et al., 2002). The nucleation rate in the simulation with the biogenic nucleation mechanism is higher in the boundary layer compared to the Bn simulation

because of abundant monoterpene, but still, at around 0.03 cm$^{-3}$ s$^{-1}$, too low to produce frequent NPF events. The rate then decreases until 2 km in altitude, where it starts to increase with height until 14 km. When we decrease the monoterpenes oxidation rate and increase the monoterpenes emission rate (from Bio to BioOxEm), the NPF rate increases by up to a factor 160 and growth rates increase by a factor of 11 in the UT because more monoterpenes is transported to the UT. Averaged over all heights, the nucleation rates from these three biogenic simulations are factors of 160 to 200 larger than in the Bn and Bn×10

simulations.

The growth rate for the biogenic nucleation between 1.7 nm and 3 nm in diameter is driven by the concentrations of the condensable gases. For binary nucleation, which is driven by $H_2SO_4$, there is no consideration of the growth rate between



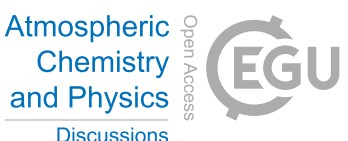

1.7 nm and 3 nm in the model calculations and therefore it is not shown in Fig. 5. The growth rates in the simulations with biogenic nucleation (Bio, BioOx, and BioOxEm) decrease with height because the concentration of HOMs decreases by a

factor of around 1000 from the surface to 14 km in all simulations.

The differences in the nucleation and growth rates between the binary and biogenic nucleation mechanisms are generally reflected in the aerosol number concentrations (Fig. 5 c, d, and e). Following the nucleation and growth rates, the nucleation mode aerosol number concentrations are very low below 4 km in all simulations. Above 4 km the concentration increases with height. The differences in nucleation mode aerosol number concentration between the two simulations with binary nucleation

are small due to similar nucleation rates, except for those between 10 and 16 km with a maximum enhancement of 73 % for Bn×10 compared to the default Bn simulation. With biogenic nucleation (Bio, BioOx, and BioOxEm), the nucleation mode concentration peaks at around 14 km ($25000\ \mathrm{cm^{-3}}$). Comparing the simulations with the most intensive nucleation (BioOxEm) to the standard biogenic nucleation simulation (Bio), we find that the 1800 times higher nucleation rate and 10 times higher growth rate result in a factor of 18 higher nucleation mode concentration at 14 km.

The Aitken mode profiles in the simulations with binary nucleation mechanisms (Bn and Bn×10) have two peaks at around 8 km and 14 km. The concentrations in Bn and Bn×10 simulations are similar except for between 12 and 16 km in altitude where the difference is likely due to the higher nucleation mode aerosol concentrations. The simulations with biogenic nucleation mechanisms (Bio, BioOx and BioOxEm) also have two peaks at 5 km and 14 km. The concentrations in those peaks are 11 times higher in the BioOxEm than in the default Bio simulation. Interestingly, there is no corresponding peak in the nucleation

and growth rates at 5 km in any of the three biogenic nucleation simulations. We find that the Aitken mode peaks at 5 km are due to transport from outside of the regional domain (i.e., from the global model) from the same altitudes where nucleation rate is greater than $0.1\ \mathrm{cm^{-3}\ s^{-1}}$ (Supplementary Fig. A2 and A3). Whether particles are formed within or outside the regional domain is investigated in more detail in Sect. 3.4.

The accumulation mode aerosol number concentrations are greatest below 2 km (on average between $500\text{-}2500\ \mathrm{cm^{-3}}$) in all

simulations. They quickly decrease to almost-zero between 6 and 12 km in altitude above which the concentrations increase again to form a peak at around 14-15 km. The BioOxEm simulation has more accumulation mode aerosols below 2 km than the BioOx simulations even though the BioOxEm simulation has fewer Aitken mode particles to grow from in the regional domain. This suggests that the Aitken mode particles are not growing into the accumulation mode size range within the regional domain, but rather in the global domain and are then transported into the regional domain (Supplementary Fig. A2). The peak

in accumulation mode number concentration at around 14-15 km is also associated with peaks in nucleation and Aitken mode concentrations, implying that the newly nucleated particles can grow to larger sizes in the UT.

In the boundary layer both the binary and biogenic nucleation mechanisms produce similar particle number concentrations in their default scenarios (Bn and Bio). When we change the monoterpenes oxidation and emission rates (BioOx and BioOxEm), aerosol number concentrations increase by factors of 3-5, especially for Aitken and accumulation mode aerosols. This suggests





that the aerosol concentrations are very sensitive to the representation of biogenic nucleation in the boundary layer. Conversely, the lack of binary nucleation in the boundary layer means that the details of the binary nucleation process are not important for the boundary layer.

### 3.3  Cloud condensation sink

The condensation sink suppresses NPF and models often calculate it using the aerosol surface area. We further add a condensa-
tion sink due to cloud droplets and ice crystals to suppress in-cloud NPF in the global and regional model domains (Kazil et al., 2011). It is applied to the simulations with biogenic nucleation (BioOx and BioOxEm) to produce the simulations BioOxCCS and BioOxEmCCS. In the UKCA model, the typical aerosol condensation sink varies between 0.003 to around 0.01 s$^{-1}$ over all heights, with a maximum domain average of 0.04 s$^{-1}$. After adding the condensation sink from cloud droplets and ice crystals, the overall condensation sink is doubled.


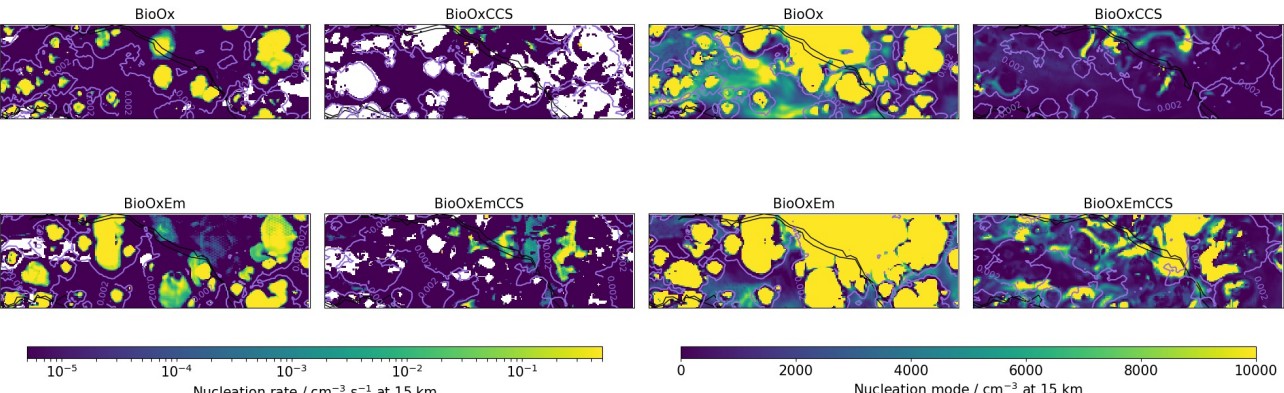

**Figure 6.** Maps of regional domain nucleation rate (left) and nucleation mode aerosol number concentrations (right) in the simulations BioOx, BioOxCCS, BioOxEm, and BioOxEmCCS at a height of 15 km and at 16 UTC on 17 September 2014. Contours highlight the locations of clouds and are drawn where the cloud water content is equal to 0.002 g kg$^{-1}$. The white areas in the nucleation rate maps have zero values and cannot be specified by a log-scale plot.

The addition of a cloud condensation sink substantially alters the spatial distribution of the nucleation rates and particle con-centrations. Figure 6 shows that adding the cloud condensation sink almost completely suppresses NPF in the cloudy regions, which is evident from the holes in the spatial pattern of nucleation rate with rates lower than $10^{-5}$ cm$^{-3}$ s$^{-1}$ at 15 km. Conse-quently the addition of the cloud condensation sink results in lower nucleation and Aitken mode particle concentrations (Fig.
7). NPF continues to occur in the non-cloudy regions because the upward-transported monoterpenes continue to be oxidised after the clouds evaporate, especially in the simulations with reduced oxidation rates. Holes in the NPF spatial distribution also occur in the BioOxEmCCS simulation (with both reduced monoterpenes oxidation and enhanced emissions). However, these





empty areas do not cover the full extent of the clouds as they do in the BioOxCCS simulation. In the cloud outflow regions NPF rates reach $1\,\mathrm{cm}^{-3}\,\mathrm{s}^{-1}$ in the BioOxEmCCS simulation.

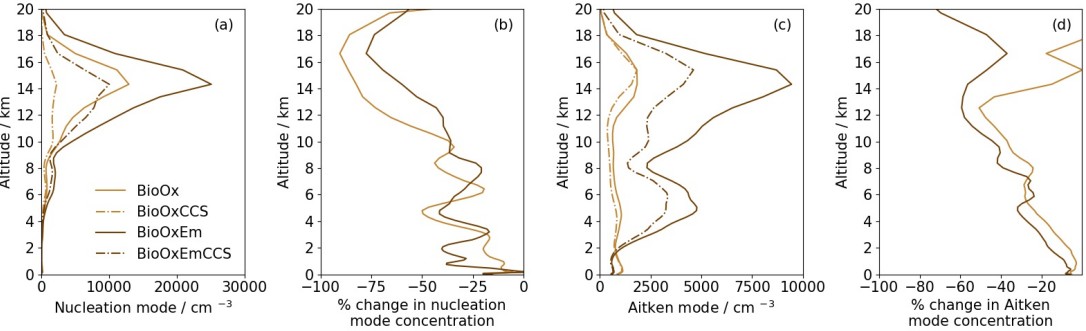

**Figure 7.** Regional domain averaged profiles from the BioOx, BioOxCCS, BioOxEm, and BioOxEmCCS simulations. Shown are the nucleation and Aitken mode aerosol number concentrations (a and c) and the percentage changes in nucleation and Aitken mode aerosol number concentrations due to the introduction of the cloud condensation sink (b and d).

## 3.4 Contribution of NPF to low-level regional particles

We now aim to quantify the number of aerosol particles in the regional domain that are formed due to NPF and growth occurring within the regional domain compared to those transported into the domain from the rest of the world. Thus, we examine the simulations where NPF is switched off in both the regional and global domains (off_allNPF) and only in the regional domain (off_regNPF) using the BioOxEmCCS as the baseline simulation. The percentage change is calculated as $100 \times (\mathrm{BioOxEmCCS} - \mathrm{off\_XXNPF})/\mathrm{BioOxEmCCS}$, where off_XXNPF denotes the simulation with either NPF switched off in both models (off_allNPF), or NPF switched off in the regional model (off_regNPF) only.





**Figure 8.** Time series of the regional domain averaged ambient aerosol number concentration profiles in the baseline BioOxEmCCS run minus those from a run in which nucleation is switched off in both the regional and global model (off_allNPF, left column) and minus those in which it is switched off in only the regional model (off_regNPF, middle column). The right column shows the time-averaged profiles and the small panel embedded in the nucleation mode aerosol number concentration profiles (top right) shows details in the lowest 3 km in altitude. Values are shown for the nucleation mode aerosol (upper row), Aitken mode aerosol (middle row), and accumulation mode aerosol (bottom row). Note that the regional domain averaged nucleation mode aerosol number concentrations in the simulation off_allNPF (top right plot) are all zero.

The time series of the aerosol vertical profiles in Fig. 8 show how number concentrations in the nucleation, Aitken and accumulation modes change when nucleation is switched off. The number concentrations of particles of all sizes are often





reduced greatly compared to the baseline simulation by switching off NPF regionally and globally. Reductions are smaller when switching off NPF in just the regional model (off_regNPF), particularly below about 14 km. The changes in aerosol number concentrations in the regional model mostly occur between 10 and 18 km. In contrast, the changes in aerosol number concentrations are large at all heights when we switch off nucleation in both models (off_allNPF).

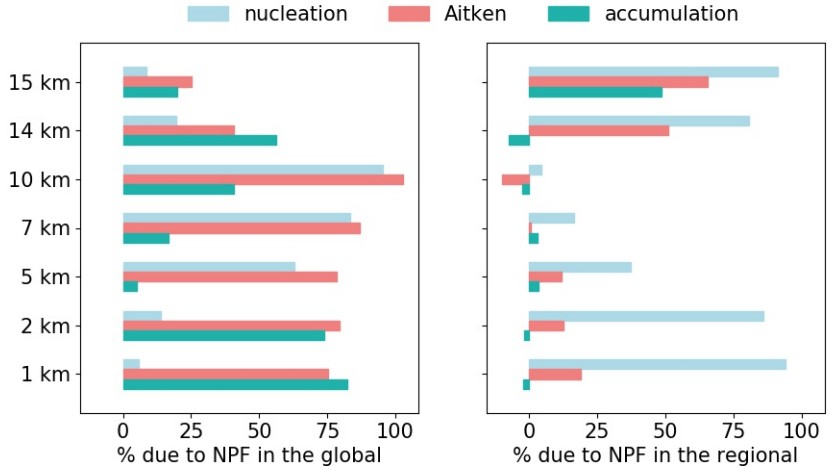

**Figure 9.** Percentage contribution of NPF to the nucleation, Aitken and accumulation mode aerosol number concentrations in the regional domain from the global model (left) and regional model (right) at various altitudes.

The percentage contribution of NPF to aerosol concentrations varies with the particle size and altitude (Fig. 9). The larger the
particle size, the smaller the influence from NPF occurring in the regional model.

The nucleation mode particle concentration is dominated by NPF in the regional model for all heights except 3-10 km, with contributions of more than 80 % above 10 km in altitude. It demonstrates that formation of nucleation mode aerosol occurs on relatively short time scales and hence also small spatial scales. These results show that the increased nucleation mode aerosol concentrations seen in the UT in Fig. 5 when switching from the baseline biogenic nucleation (Bio) to the enhanced biogenic
nucleation schemes (BioOx and BioOxEm) are mainly caused by additional biogenic NPF within the regional domain rather than outside of it. At 3-10 km, NPF in the regional model contributes to less than 45 % of the nucleation mode concentration, so 55 % is formed in the global model and advected in.

The effect of advection of nucleation mode aerosol into the regional domain at different altitudes is determined partly by the different vertical profiles in the two domains. Nucleation rates in the global model at 3-10 km are on average 25 times greater
than in the regional model in the BioOxEmCCS simulation (Supplementary Fig. A3). The smaller nucleation rate in the regional model is likely due to the higher condensation sink generated by explicit cloud convection, and by the different vertical profiles of trace gases caused by resolved convection. We aim to investigate this in future studies. For example, at around 8 km where the nucleation mode concentration is about a factor of 4 higher than in the regional model, the regional condensation sink is





about a factor of 3 higher than in the global model, while the concentrations of monoterpenes are within 10 % (Supplementary

Fig. A3). The higher condensation sink results in around 50 times lower nucleation rate in the regional model. These numbers suggest that nucleation rate in the global model is higher than in the regional model in this deep convective environment due to the global model failing to resolve the small-scale spatial variations in trace gases, aerosols and clouds.

The percentage contribution of NPF to the Aitken mode particle concentration in the regional model is also dominant above 14 km in altitude, but is small below that height. Below 2 km (in the boundary layer), it is around 12 %-19 %. Between 14 and

15 km in altitude, around 51 %-66 % of the domain-averaged Aitken mode concentration is from NPF in the regional domain, and 25 %-41 % is from the global model NPF. The percentages do not sum to 100 % because of the contribution from primary particles. At 5 km, NPF in the regional model accounts for 12 % and outside the regional domain, the global model accounts for 78 % of the Aitken mode concentration. This result supports the arguments that the extra Aitken mode aerosol that we saw in Fig. 5 when switching from BioOx to BioOxEm at 5 km in altitude were due to NPF in the global model (either at 5 km,

or at other heights followed by vertical transport). The overall percentage contribution of NPF to Aitken mode aerosol in the regional domain is smaller than that of the nucleation mode aerosol because forming the Aitken mode aerosol requires a longer time and is affected by coagulation and scavenging.

The accumulation mode aerosol is the least dependent on NPF from the regional model. Above 15 km, the contribution of NPF in the regional domain to the accumulation mode aerosol is 49 % and 20 % is from global model NPF, meaning that the

regional model is able to form some accumulation mode aerosol via NPF in the time available in the domain. In Fig. 5 we saw more accumulation mode aerosol below 2 km in altitude as the biogenic nucleation rates were increased (from Bio to BioOx and from BioOx to BioOxEm). Fig. 9 confirms that NPF in the regional domain does not lead to the formation of the additional accumulation mode below 2 km because NPF actually slightly reduces the concentrations, showing that the regionally-formed Aitken mode particles do not grow to accumulation mode sizes at these heights. A few possible explanations for the slight

reduction in concentrations due to NPF below 2 km are: that NPF causes the aerosol size distribution to shift to a smaller size as was reported in Sullivan et al. (2018); increased precipitation removes more accumulation mode aerosol (Supplementary Fig. A9 and A11); due to upward transport from the surface to higher altitudes; it could also be caused by the randomness of a different convection field. We further investigate the issue of aerosol vertical transport in Sect. 3.5.

Overall, our findings in this section show that in the regional domain below 2 km, Aitken and accumulation mode particles are

dominated by NPF occurring outside of the 1000 km regional domain and in our study, these particles come from the global model. It implies that the boundary layer CCN, which influence cloud droplet number concentrations, are originally transported from outside the domain.

## 3.5 Convective transport of particles

Section 3.4 showed that NPF in the regional domain produces only around 10-20 % of Aitken mode aerosol in the boundary

layer and has a negligible effect on the accumulation mode. Here we aim to understand why these regionally nucleated particles





have a weak effect on boundary layer Aitken and accumulation mode particles. We examine five simulations in which NPF in the regional model is only allowed at specific altitudes (1-4 km, 4-7 km, 7-10 km, 10-13 km, and 13-16 km). Percentage differences are calculated using $100 \times (\mathrm{NPF\_XXkm} - \mathrm{off\_regNPF})/\mathrm{NPF\_XXkm}$, where $\mathrm{NPF\_XXkm}$ denotes one of the simulations with NPF switched on only between 1-4 km, 4-7 km, 7-10 km, 10-13 km, and 13-16 km.

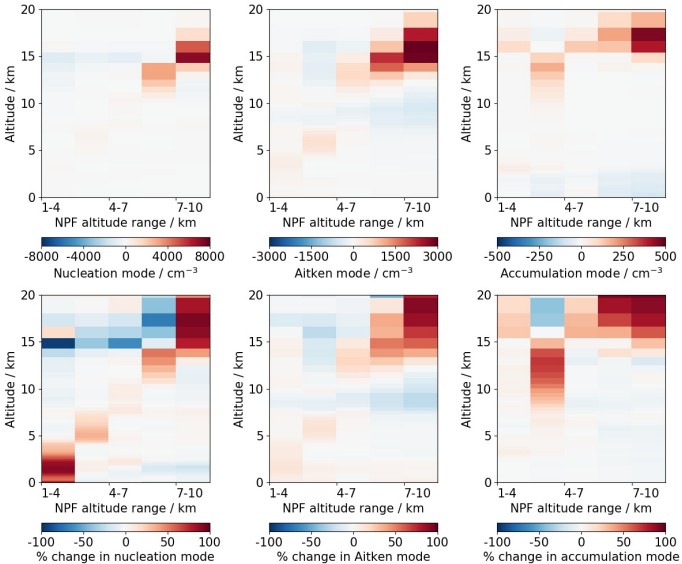

**Figure 10.** The absolute (upper row) and percentage (lower row) changes of the regional domain averaged profiles of ambient number concentrations of (left column) nucleation mode aerosol, (middle column) Aitken mode aerosol, and (right column) accumulation mode aerosol, between the simulations with NPF switched on at certain altitudes (NPF_1-4km, NPF_4-7km, NPF_7-10km, NPF_10-13km, and NPF_13-16km) and the simulation with no NPF in the regional domain (off_regNPF).

Figure 10 shows the absolute and percentage domain-average effects of NPF occurring in these altitude layers. The absolute differences are most significant for the nucleation and Aitken mode above 10 km, and the accumulation mode above 15 km in altitude. NPF above 10 km (in the UT; NPF_10-13km and NPF_13-16km) perturbs particle concentrations almost entirely at or above the heights where it occurs. For example, in the NPF_13-16km simulation, the nucleation mode concentration changes by around 7200 cm$^{-3}$ (79 %) at these altitudes, the Aitken mode concentration changes by 3900 cm$^{-3}$ (70 %), and accumulation mode concentration changes by 470 cm$^{-3}$ (83 %). It again confirms that the regionally-formed nucleation mode particles grow and coagulate to form Aitken and accumulation mode aerosol within the domain. NPF between 13 and 16 km (NPF_13-16km) contributes to nucleation and Aitken mode particles between 12 and 20 km and to accumulation mode particles between 13 and 20 km. The vertical extent over which the perturbations occur implies that nucleation and Aitken mode particles are transported mostly upwards (but also downwards) from the altitudes where NPF takes place, while the accumulation mode particles in most of the time are only transported upwards.

NPF in the regional model UT contributes very little to particle concentrations below 2 km in the regional domain during the





3-day dry season simulation. The contributions of NPF above 10 km to particles below 2 km are -0.3 cm$^{-3}$ (-0.01 %) for the nucleation mode, 33 cm$^{-3}$ (1.1 %) for Aitken mode and -126 cm$^{-3}$ (-4.2 %) for the accumulation mode. In cloudy downdrafts below 2 km, Aitken mode concentrations resulting from NPF above 10 km occasionally reach a maximum of 100 cm$^{-3}$ (but

on average, it only accounts for 0.13 % of the regional-domain concentration of all times) and a maximum of 60 cm$^{-3}$ (on average contributing to 0.08 % of the domain particles) for the accumulation mode. It shows that deep convection can transport particles that are formed in the regional UT to low altitudes when convective downdrafts are strong. However, these number concentrations have a negligible effect on the domain-mean number concentrations below 2 km because deep convection covers only around 4 % of the domain below 2 km. Thus, even though NPF above 10 km in the regional model can form the Aitken

and accumulation mode particles within the domain, the majority of the particles either stay in the UT or leave the domain by horizontal transport. These particles may be transported downwards on larger spatial scales, but not on the scale of ∼1000 km simulated here.

NPF below 10 km produces fewer particles than NPF in the UT. NPF between 7 and 10 km causes a peak in the Aitken mode concentration up to around 590 cm$^{-3}$ (20 %) between 11 and 15 km, while there is no increase in Aitken mode in the 7-10

km height range where NPF is occurring. It shows that Aitken mode particles at 11-15 km are affected by ascent of nucleation mode aerosol from lower altitudes followed by growth to Aitken mode sizes. NPF in all altitude layers contributes to accumulation mode particles in the 10-15.5 km layer, with domain- and time-mean increases as large as 180 cm$^{-3}$ (65 %). Overall, these results show that NPF below the UT can contribute to Aitken and accumulation mode particles in the UT.

The addition of NPF in the regional domain *reduces* the accumulation mode number concentrations below 4 km due to en-

hancement of rain rates by 1.3-2.6×10$^{-6}$ kg m$^{-2}$ s$^{-1}$ (2.8 %-5.3 %) associated with additional particles formed by NPF, which is similar to what was seen in Sect. 3.4 (Supplementary Fig. A5, A6, A7 and A8). One possibility is that an increased aerosol number concentration by NPF results in a higher cloud and ice content and subsequently leads to more rain which removes low-level aerosols. The mechanisms of the overall enhanced scavenging are different when we switch on the NPF above and below 10 km in the regional domain. Supplementary Fig. A9, A10, A11, and A12 show that NPF above 10 km causes increases

in cloud water and ice content, whereas NPF below 10 km results in a strong increase in cloud water along with an overall decrease in ice content, but these increases could also be due to model randomness.



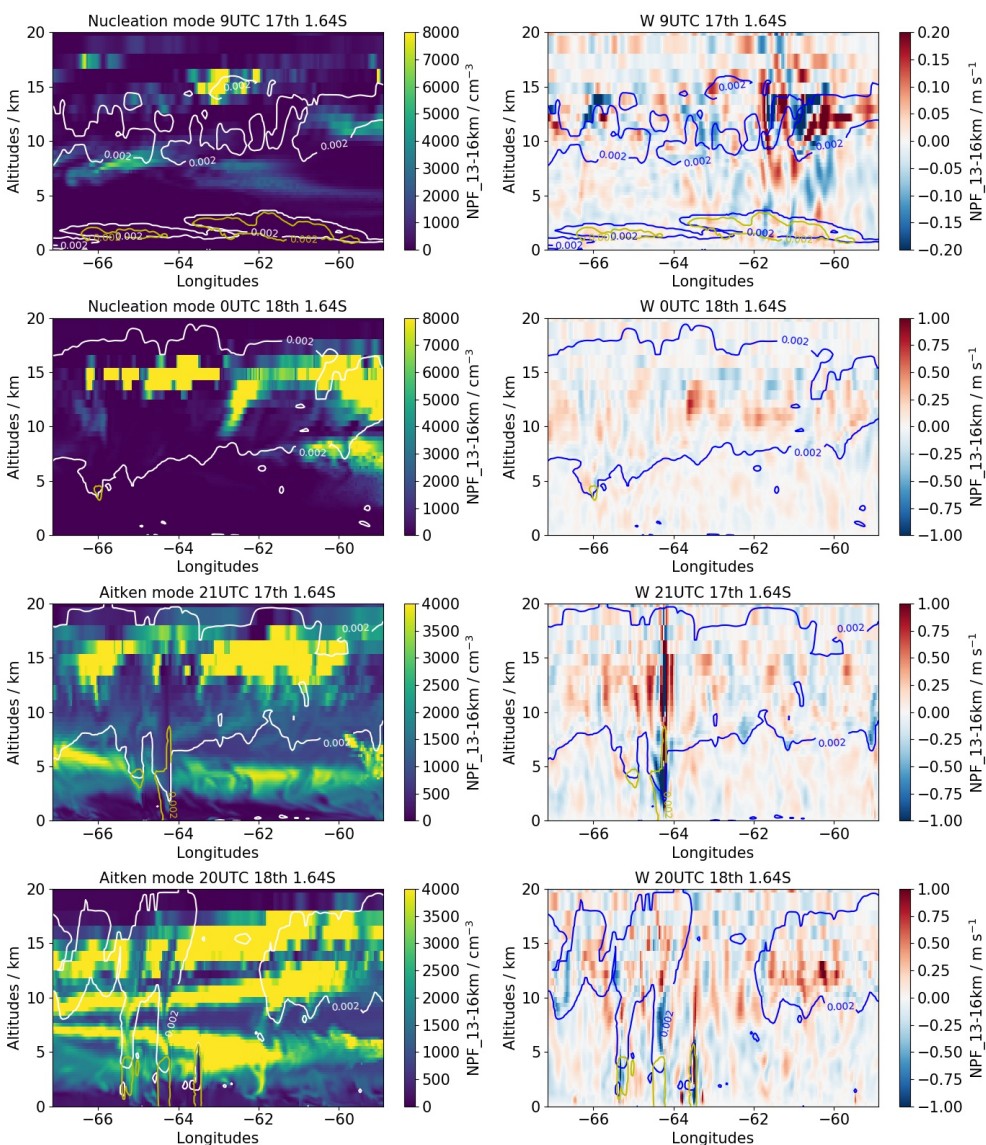

**Figure 11.** West to east vertical slices at 1.64 S of nucleation mode (first two rows), Aitken mode (lower two rows) number concentrations and the vertical velocity from the NPF_13-16km regional simulation at various times (see panel titles) in order to highlight vertical transport. Easterly winds were dominant. White and blue contours highlight clouds (liquid plus frozen water content = 0.002 g kg$^{-1}$) and yellow contours denote rain mass mixing ratios of 0.002 g kg$^{-1}$. Videos of the nucleation, Aitken and accumulation mode aerosol slices can be found in the supplementary.

Figure 11 shows the vertical slices of nucleation mode, Aitken mode particles and vertical velocity that exemplify the vertical transport. A plume of nucleation mode aerosol at 9 UTC on 17 September 2014 descends from around 10 km to 7 km between 64 W and 66 W and quickly exits the regional domain to the west (see videos of vertical transport in supplementary). Similarly,





at 0 UTC on 18 September between 62 W and 63 W a 'finger' of nucleation mode aerosol extends from the UT down to around 9 km and is then diluted within three hours. The two slices also show clear downward transport of Aitken mode aerosol from around 7 km to below 2 km in altitude between 64 W and 66 W, which is associated with cloud (white contours). The downward motion of Aitken mode aerosol is a potential explanation for the excess accumulation mode seen below 2 km in BioOxEm compared to BioOx in Fig. 5 e (Sect. 3.2), namely that the additional nucleation in BioOxEm leads to more

nucleation and Aitken mode aerosols in the global domain, which enters the regional domain below 7 km from the global model and those Aitken mode formed below 7 km within the regional domain, are transported downwards into the boundary layer by convection and then grows to accumulation mode sizes. The Aitken mode aerosol from the global model nucleation are mostly formed above around 2 km in altitude where nucleation rates start to significantly affect aerosol concentrations and grow to accumulation mode as they sink into lower altitudes (Supplementary Fig. A3 and A4).

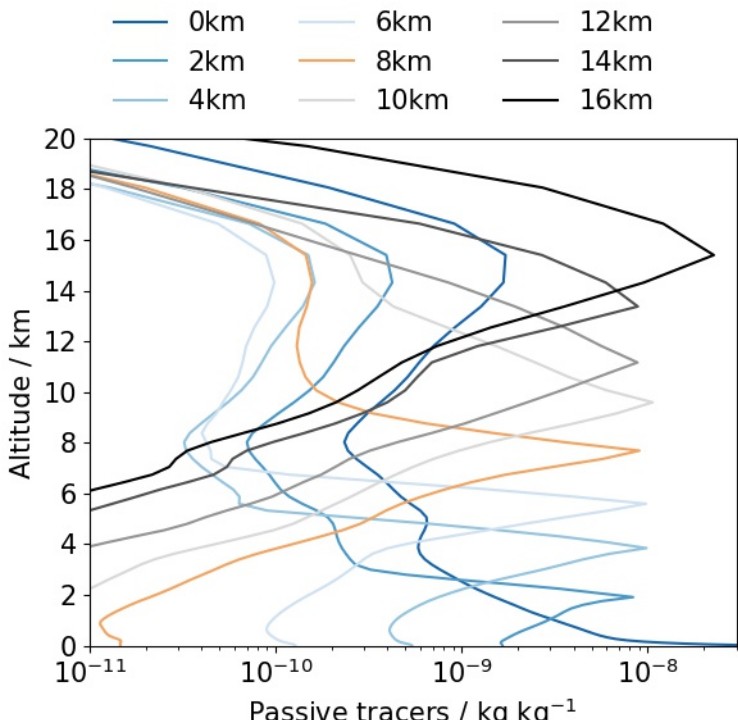

**Figure 12.** Regional domain averaged profiles of passive tracer mass mixing ratios emitted from 9 different model levels with vertical thicknesses ranging from 16 m at the surface (blue) to 2000 m at 16 km (black) in altitudes.





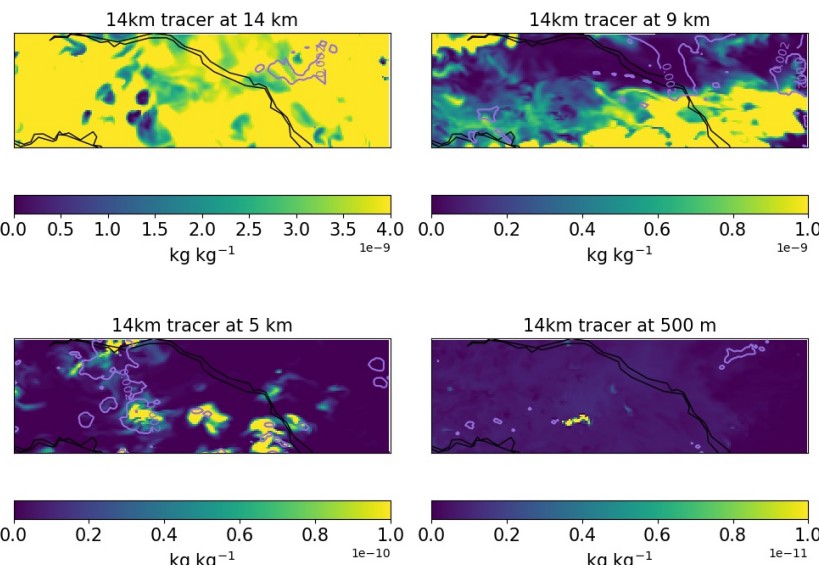

**Figure 13.** Maps of the mixing ratio of a passive tracer emitted at 13-16 km altitude. Mixing ratios are shown at 14 km, 9 km, 5 km, and 500 m at 21 UTC on 17 September 2014 (47 hours after the first release of the tracer). The upper limits in the four maps are different.

We also implemented transport-only passive tracers in the model to understand the vertical transport efficiency of air in the convective environment. The tracers were emitted within 9 vertical model layers using the same constant emission rate as the default monoterpenes emission rate from CMIP5 inventories with no deposition. Figure 12 shows the domain averaged tracer mixing ratios for the whole simulation period. Figure 13 also shows maps of the instantaneous mixing ratio of the 14 km altitude tracer at different altitudes. The 9 tracers have the largest concentrations where they were emitted, but only those that

are emitted below 6 km reach the surface. The tracers emitted at higher altitudes are redistributed both up- and downwards by around 5 km, creating bands with depths between 8 and 10 km. An example can be seen from the maps of tracer that is emitted at 14 km in altitude (Fig. 13). Similar scales of vertical mixing of aerosols can also be seen from global model (Supplementary Fig. A4). We compare the tracer mixing ratios at 9 km, 5 km and 500 m to 14 km in altitude in order to obtain the transport ability. We find that on average less than 5 % of the 14 km tracer reaches an altitude of 9 km, 0.13 % reaches 5 km, and barely

any is transported to 500 m in altitude (0.01 %). Consequently, over a 3-day period a convective environment of dimension 1000 km can transport air downward in sufficient quantities to significantly affect the domain mean by at most 5 to 8 km within the regional domain, but the influence is less than a few percent. The Amazonia in East-West direction is around a factor of 3 of the size of our regional domain. Therefore, if air masses keep the descending motion, the number of particles being transported into the boundary layer would be expected to increase with a greater domain. A regional simulation with a higher resolution

would be likely to be more efficient at transporting aerosol vertically but we did not perform such a simulation in this paper in order to keep a reasonably large domain. In contrast, the model exhibits strong upward transport which allows substantial amount of tracers, especially for those emitted below 4 km to reach 16 km in altitude. Other tracers are transported upward by





~0.5-4 km. Therefore, in the regional domain, the aerosol can be transported upward by as far as 16 km, but downward by at most 8 km.

## 4 Discussion and Conclusions

We used a global model with a 4 km-resolution nested regional domain (of size 1080 km by 440 km) to study the influence of deep convection on new particle formation (NPF) and the budget of cloud-forming aerosol particles in Amazonian boundary layer.

The regional-scale simulations show that deep convection regulates the vertical distribution of trace gases and aerosol particles by efficiently transporting monoterpenes from the surface to the UT. In the UT, monoterpenes can be oxidised within a few hours, and with low temperature and condensation sink, new particles are efficiently formed. Consistent with observations (Andreae et al., 2018; Williamson et al., 2019) and global model simulations (Pierce and Adams, 2007; Merikanto et al., 2009; Wang and Penner, 2009; Dunne et al., 2016), our regional simulations of a convective environment show that NPF is strongest in the UT, leading to the greatest number concentrations of nucleation and Aitken mode particles (a total of more than 10000 $cm^{-3}$).

The rate of NPF in the UT is reduced and spatially strongly modulated by the condensation sink of trace gases and nuclei on cloud droplets and ice particles. When this additional 'cloud condensation sink' is included in our regional model, mean concentrations of nucleation and Aitken mode particles in the UT are reduced by 50 %. The formation of particles primarily in detraining convective clouds is consistent with several observations (Clarke et al., 1998; Twohy et al., 2002; Andreae et al., 2018; Williamson et al., 2019). This localised cloud sink is straightforward to include in a convection-resolving model, but would be more difficult to include in a global model in which clear and cloudy air parcels in the UT are not explicitly simulated.

The typical vertical profiles of nucleation mode and Aitken mode particles, with peak concentrations in the UT, are created through NPF in the regional model on the timescale of a few days. With typical easterly winds in the area of Amazonia that we studied, the nucleation and Aitken mode particle profiles in the UT are therefore created on spatial scale of a few hundred kilometres as air advects across the rainforest. However, below the UT the environmental conditions required to create the nucleation and Aitken mode profiles are not ideal in the regional domain. The regional influence of NPF on the accumulation mode particle profile is important only at the highest altitudes in the UT ($\sim 15$ km) and is negligible at lower altitudes. Similarly, the influence of regional-domain NPF on Aitken mode aerosols is significant in the UT, but NPF accounts for only around 10-20 % of particle concentrations in the boundary layer. This weak effect is because of the longer time taken to form the larger particles following NPF, which means that particles formed by NPF above the boundary layer are advected out of the domain before they reach the larger sizes or before they can be transported downwards.

Below approximately 10 km altitude, the regional model simulations show that nucleation and Aitken mode particles are not substantially affected by NPF on the timescale of 3 days. In our regional domain of size $\sim 1000$ km aligned with the mean





easterly wind, aerosol in the boundary layer is mostly produced outside the region (in the global model) and advected into the
domain below 10 km altitude. Consistent with previous global model studies, we find that these advected particles were mostly
formed by NPF, but on much larger spatial scales than the 1000 km domain we simulated.

NPF is strongest above 10 km altitude, but our simulations show that it can affect particles below this altitude through vertical
transport in the deep convective environment. The regional simulations show clear plumes of particles being transported in
downdrafts, and these have been observed Wang et al. (2016). However, extremely few particles formed above 10 km altitude
are transported all the way to the boundary layer (less than 1 %) during the 3 days of our regional simulation. Our simulation
(BioOxEmCCS simulation) overestimates the particle concentrations in the boundary layer by less than a factor of 3. Therefore,
the percentage of particles by convection transport are expected to be greater if the boundary layer is simulated to be less
polluted, but the contribution will still be small. Rather, the simulations show that NPF-formed aerosol above 10 km altitude
is transported only a few kilometres downwards (to around 8-10 km in altitude), while the aerosol entering the boundary layer
originated from altitudes below 7 km. This lower-altitude aerosol was not formed by NPF during the 3-day simulation, but
was advected into the regional domain and then transported downwards. Figure 14 shows a sketch of this vertical transport.
This limited vertical transport, especially for Aitken and accumulation mode, is because the downdraft within one convective
cycle in the regional domain is not strong enough to bring a large number of particles from the UT down to the boundary layer,
not even for the passive tracers that are not scavenged during vertical transport. These results show that new aerosols that are
formed within a 1080 km by 440 km regional domain in the Amazonian dry-to-wet season are not the major source of the
boundary layer aerosol particles for such a domain.





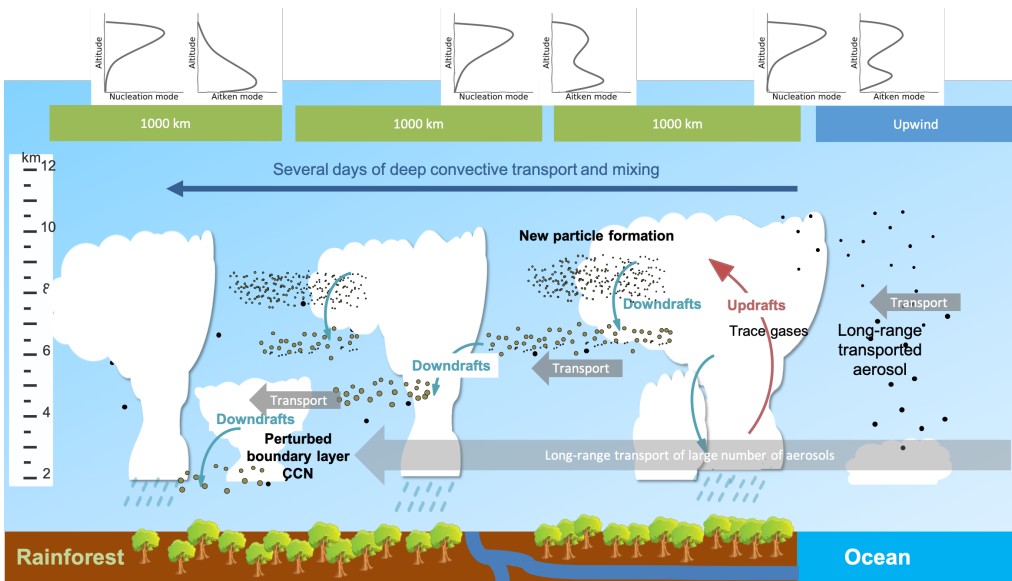

**Figure 14.** A schematic diagram of downward transport and mixing of aerosols associated with NPF. The nucleation and Aitken mode aerosol profiles in upper panels are example number concentrations in the upwind, within, and in the downwind of the simulated 1000 km by 400 km region.

Our results are similar to those of Clarke et al. (2013), who reported that particles in the free troposphere are likely to be transported from thousands of km away before they finally contribute to boundary layer CCN. The extent of aerosol vertical transport in our simulations is in line with Gerken et al. (2016) who showed that $O_3$ from 2-7 km altitude may enter the boundary layer during convective storms occurring during GoAmazon2014/5. Other observational studies (Giangrande et al., 2016; Machado et al., 2021) also reported that downdrafts occurred most frequently below the freezing level (10 km) in Amazonia and that the horizontal extent of the downdraft decreased with increasing altitude. Similarly during GoAmazon2014/5, observations in Tang et al. (2016) showed that the air exhibited downward motion between $\sim 700$ hPa and the surface during the day (10-18 local time), and above $\sim 700$ hPa during the rest of the day, but these two periods were interrupted by the upward motion, inhibiting the downward motion from the UT to the boundary layer. We do not find any significant increases in particle number concentrations below 2 km associated with convective downdrafts, which is consistent with the analysis of observations during GoAmazon2014/5 and ACRIDICON-CHUVA campaigns (Machado et al., 2021). Wang et al. (2016) reported rapid downward transport of free tropospheric aerosols that could have been formed in the cloud outflow. With ATTO tower data, Franco et al. (2022) found particles smaller than 50 nm to enter the boundary layer that was likely to be caused by gust front downdrafts or rain. Our results also show instantaneous increase of Aitken mode particles up to $100^{-3}$ associated with cloudy downdraft but only occurs occasionally. However, over the 3-day simulation of the 1000 km domain, these downward transported particles are negligible, making the downward transport efficiency small in our simulations.

In the dry-to-wet season transition, NPF in the boundary layer has a very limited effect on Aitken mode and accumulation





mode particles below 2 km in altitude. Instead, in the wet season when the environment is less polluted, the percentage con-
tribution of convective downdraft from UT to the boundary layer particles are likely to be greater. The small effect of NPF
upon boundary layer CCN may be further weakened by strong biomass burning events that took place in September 2014,
which would increase the aerosol condensation sink and suppress NPF. These events were not included in our simulations.
This additional condensation sink may be less important in the wet season when biomass burning is rarer and the condensation
sink is generally lower than the season studied here.

Our results support the conclusion of Andreae et al. (2018) that NPF in the Amazonian UT is sustained by the upward transport
of biogenic vapours in deep convection. Our simulations show that very high particle concentrations above 10 km altitude are
created within a few days of advection of air over the rainforest. However, our results show that these newly formed particles
in the UT do not contribute to boundary layer particles via vertical mixing and transport on the timescale of a few days. Our
results agree with Andreae et al. (2018) and Wang et al. (2016) that Aitken mode particles can be transported downwards from
the lower free troposphere into the boundary layer, and such downdraft events associated with convection are apparent in the
model. However, on the timescale of a few days and a spatial scale of 1000 km, such transport has a small effect on mean
particle concentrations in the boundary layer. This is even the case for passive tracers, so transport is the limiting factor, not
aerosol microphysics. Although our results are consistent with Andreae et al. (2018) in that NPF in the FT is an important
overall source of CCN in Amazonian boundary layer, our results show that these particles are formed on spatial scales much
larger than 1000 km, and not necessarily over Amazonia.

Overall, we have high confidence that, during the dry-to-wet transition season, Amazonian rainforest controls aerosol particle
concentrations in the UT, and that the observed high concentrations are produced directly within regions of deep convection
on the timescale of a few days. We have moderate confidence that particle concentrations below the UT are controlled by pro-
cesses occurring on much larger scales than 1000 km. Therefore, the concept of a cycle of trace gas vertical transport, particle
formation, and subsequent CCN transport into the boundary layer is unlikely to be a 'closed loop' over the selected region
in Amazonia which is around 1/3 of the forest in East-West direction, but is likely to be strongly influenced by advection of
aerosol into this regional domain.

There are some limitations of our simulations that would need to be overcome to confirm our conclusions. In particular, the
length of simulation time, the regional domain size and 4 km resolution may limit the generalisability of our results and our
understandings of the regional NPF to CCN link. The 3-day simulation time may only represent a short period of a year, there-
fore, it will not represent other time such as the wet season when the boundary layer is less polluted. The relatively coarse 4
km resolution may limit the extent of downward transport of aerosol from the UT to the boundary layer in distinct plumes.
Nevertheless, there are thermodynamic limits on the extent to which air can be exchanged in this way. If the domain were
larger, the particles would be allowed to grow and be transported in the domain for a longer time, then the number particles
enters the boundary layer from UT may experience moderate increases. A domain covering the whole of Amazonia would be
about 2-3 times larger in linear dimension than we used here, which would provide about 2-3 times longer for vertical mixing





assuming the same mean advection speed. This increase alone would not be sufficient to affect our conclusion that Amazonia is not a closed CCN production loop through new particle formation. Such a loop could exist in regions where air is stagnant over Amazonia or in the wet season with considerably more convection.

We strongly recommend that future regional modelling studies of Amazonian particles include a driving global model to fully capture the long-range transport of aerosol. We also recommend that the regional nests use an increased resolution and domain size; and that more chemical complexity is included. Comparing the wet and dry season would be helpful to gain a complete picture of the evolution of the particle number concentrations and size distributions in Amazonia.

*Data availability.*   The observations have been obtained from ACRIDICON-CHUVA campaign (Andreae et al., 2018). Raw model data are available through JASMIN service (http://www.jasmin.ac.uk/, last access: 5 October 2022). We have uploaded a subset of simulation data that were used to produce the figures to Zenodo (https://zenodo.org/record/7149562#.Yz6NW-zMLtM).

*Video supplement.*   The videos to show the vertical transport of nucleation, Aitken and accumulation mode aerosols in the regional model from the NPF_13-16km simulation can be accessed from below.
https://drive.google.com/file/d/1DTWCyaevqFDTk5UfkMrxDjArjaNq3-TY/view?usp=sharing
https://drive.google.com/file/d/19JnwQNFG-NjUETIpcaWD620kXGx0bGF0/view?usp=sharing
https://drive.google.com/file/d/10v5Ygb8zDje8GpoH00pbbBIC4aGpPJ_z/view?usp=sharing





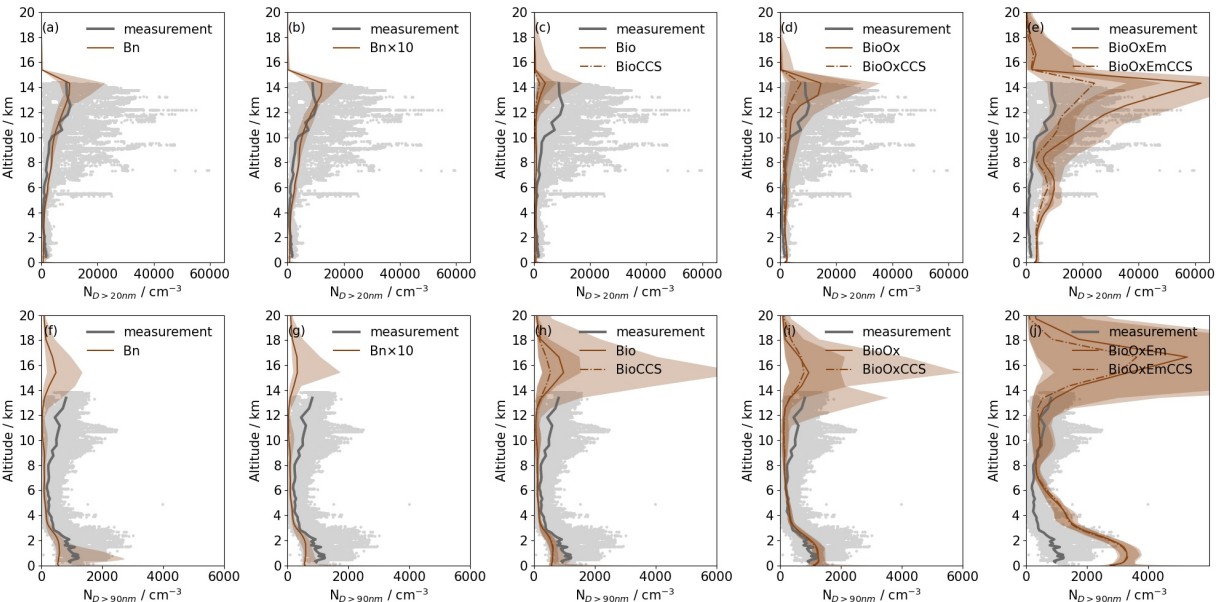

**Figure A1.** The observed and modelled vertical profiles of median number concentrations of particles with diameters >20 nm ($N_{D>20nm}$, top row) and >90 nm ($N_{D>90nm}$, bottom row). The observations are shown in dots and grey line (repeated for all panels), and the modelled results are from the various regional simulations averaged from 0 UTC on 17 September to 23 UTC on 18 September 2014, (a) Bn, (b) Bn×10, (c) Bio and BioCCS (dashed line), (d) BioOx and BioOxCCS (dashed line), and (e) BioOxEm and BioOxEmCCS (dashed line), corrected to standard temperature and pressure (Eq. 2). The shading represents 2.5 % and 97.5 % percentiles from the modelling results. The grey dots are individual observations from all flights during ACRIDICON-CHUVA with a time resolution of 1 minute, and the thick grey lines are the medians of the observations binned within the same height ranges as the regional model levels.





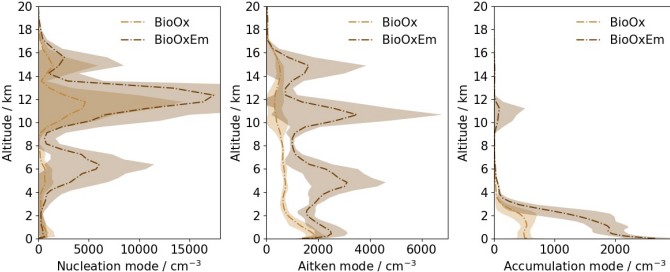

**Figure A2.** The domain averaged profiles of the nucleation, Aitken and accumulation mode aerosol in the simulations BioOx, and BioOxEm from the global model in the upwind (East) of the regional domain.





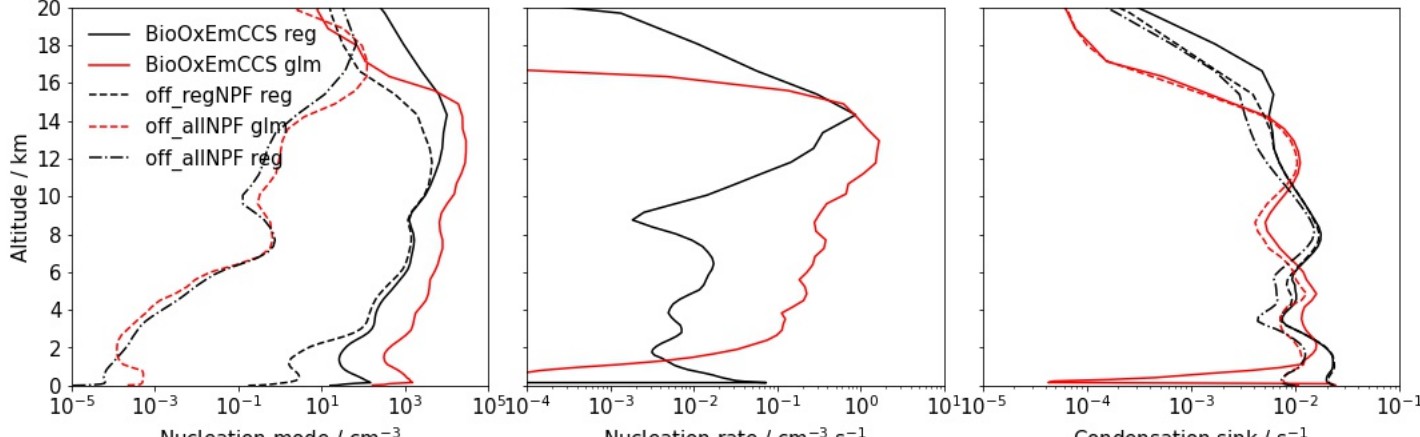

**Figure A3.** The regional domain averaged profiles of nucleation mode aerosol number concentrations, nucleation rate and condensation sink from the simulations BioOxEmCCS (solid), off_allNPF (dotted dashed), and off_regNPF (dashed) in the global model (red) and regional model (black).





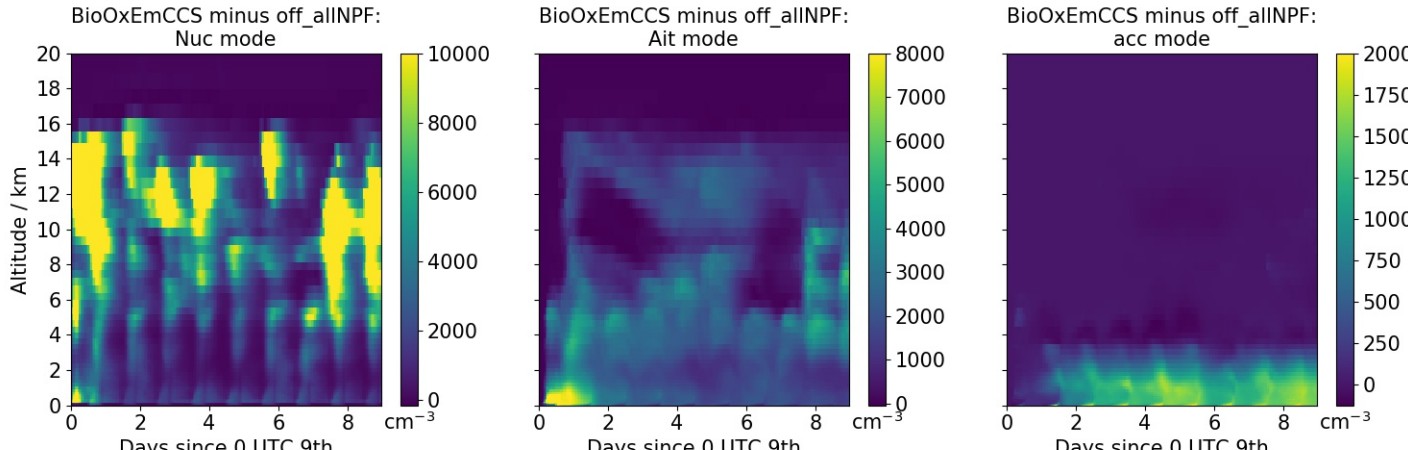

**Figure A4.** Time series of the global domain averaged aerosol number concentrations at the same location of the regional domain. The number concentrations are the differences between the BioOxEmCCS and off_allNPF simulations.





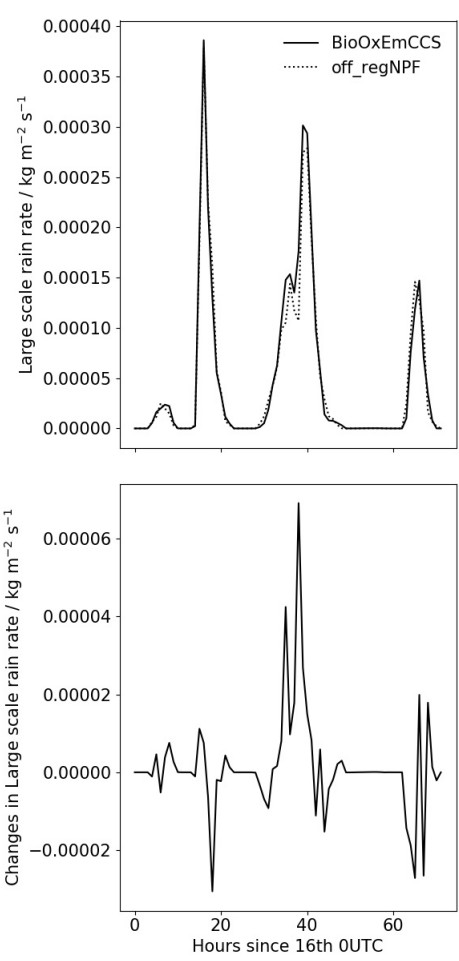

**Figure A5.** Regional domain averaged time series of large scale rain rate in the BioOxEmCCS (solid), and off_regNPF (dotted) simulations (upper panel), and the differences between NPF switched on and off (BioOxEmCCS − off_regNPF; lower panel).





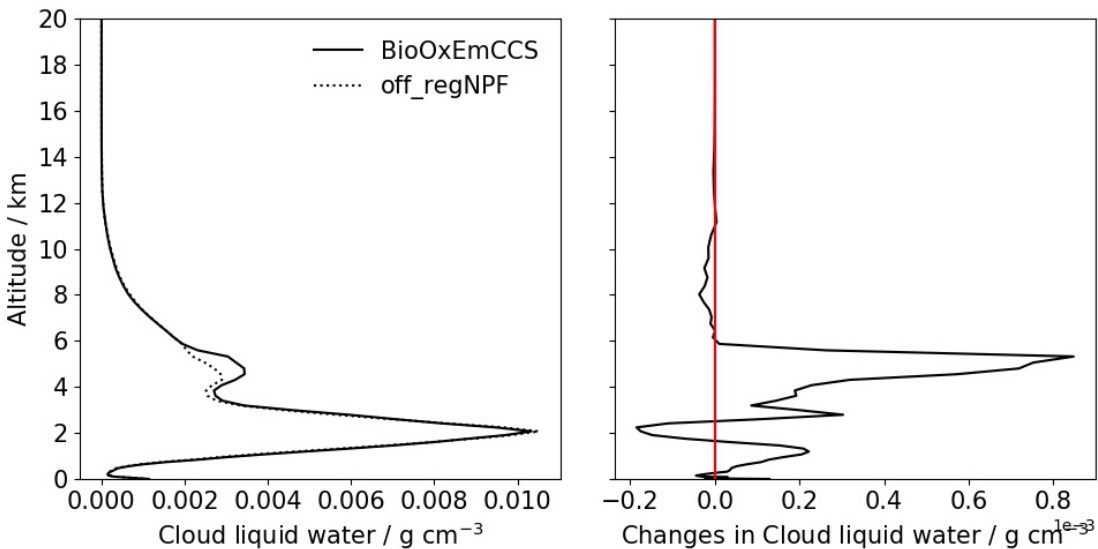

**Figure A6.** Regional domain averaged profiles of cloud liquid water content in the BioOxEmCCS (solid), and off_regNPF (dotted) simulations (left), and the differences between NPF switched on and off in the regional domain (BioOxEmCCS − off_regNPF; right).





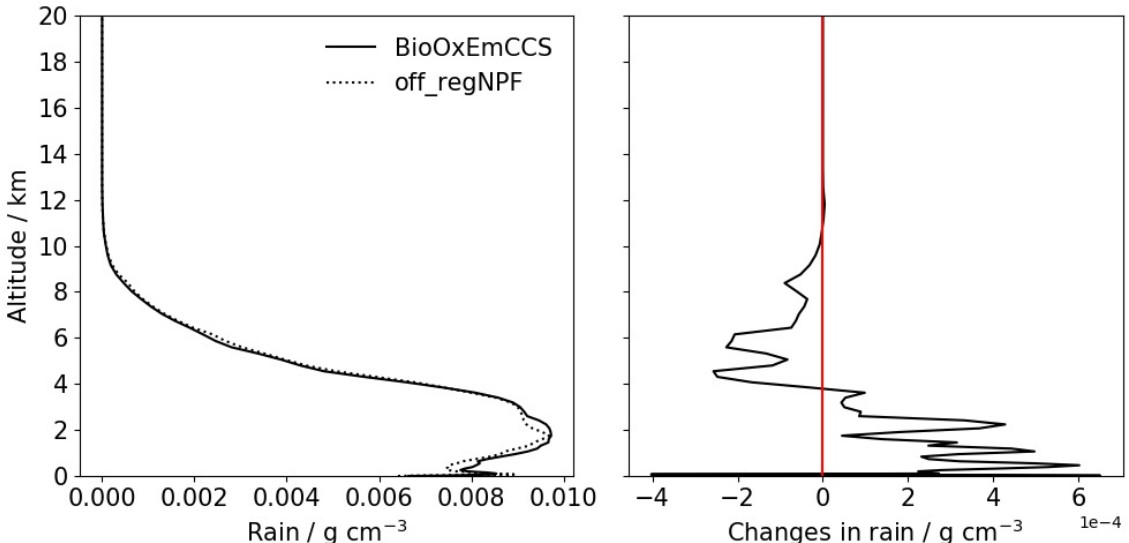

**Figure A7.** Regional domain averaged profiles of rain water contant in the BioOxEmCCS (solid), and off_regNPF (dotted) simulations (left), and the differences between NPF switched on and off in the regional domain (BioOxEmCCS − off_regNPF; right).





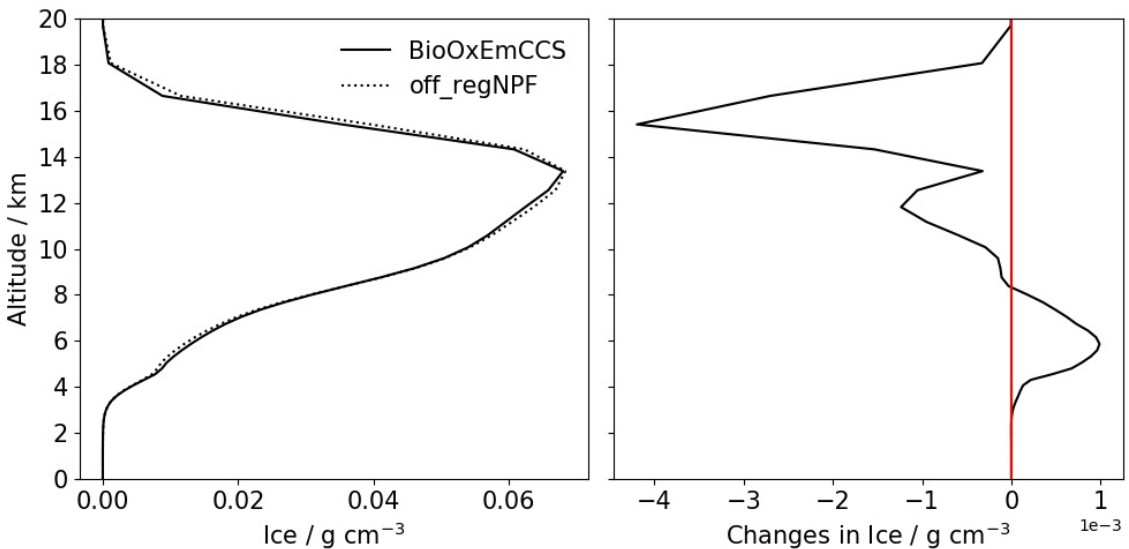

**Figure A8.** Regional domain averaged profiles of ice concentrations in the BioOxEmCCS (solid), and off_regNPF (dotted) simulations (left), and the differences between NPF switched on and off in the regional domain (BioOxEmCCS − off_regNPF; right).



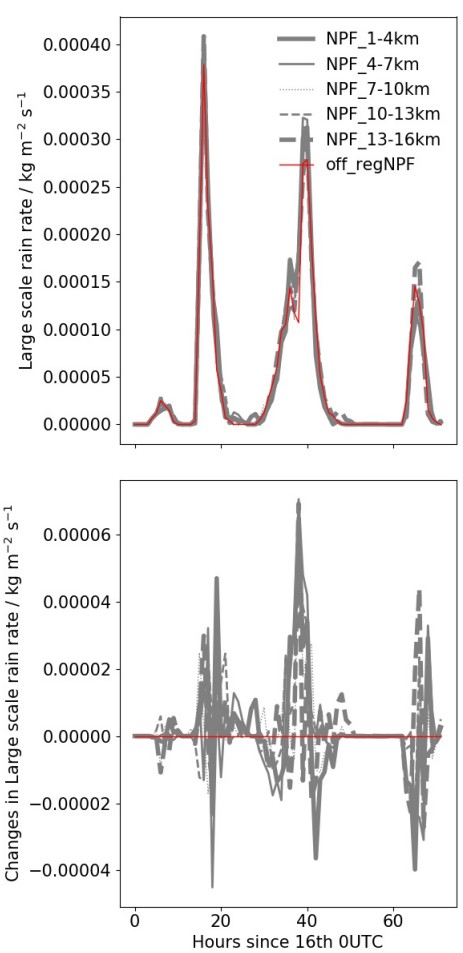

**Figure A9.** Regional domain averaged time series of large scale rain rate in the simulations NPF_1-4km, NPF_4-7km, NPF_7-10km, NPF_10-13km, NPF_13-16km, and off_regNPF (upper panel), and the differences between NPF switched on and off (NPF_Xkm − off_regNPF; lower panel).





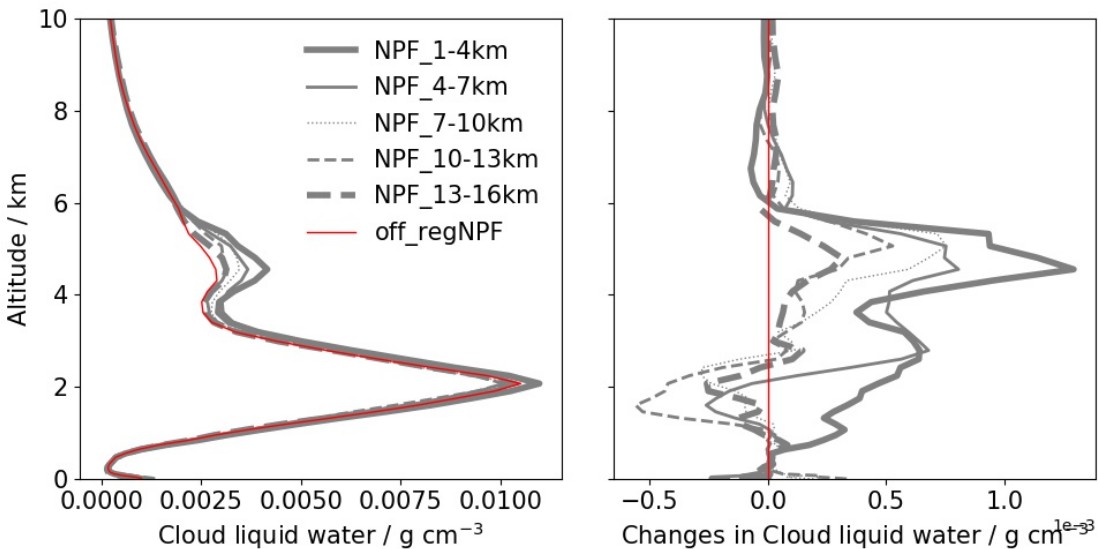

**Figure A10.** Regional domain averaged profiles of cloud liquid water content in the simulations NPF_1-4km, NPF_4-7km, NPF_7-10km, NPF_10-13km, NPF_13-16km, and off_regNPF (left), and the differences between NPF switched on and off in the regional domain (NPF_Xkm − off_regNPF; right).





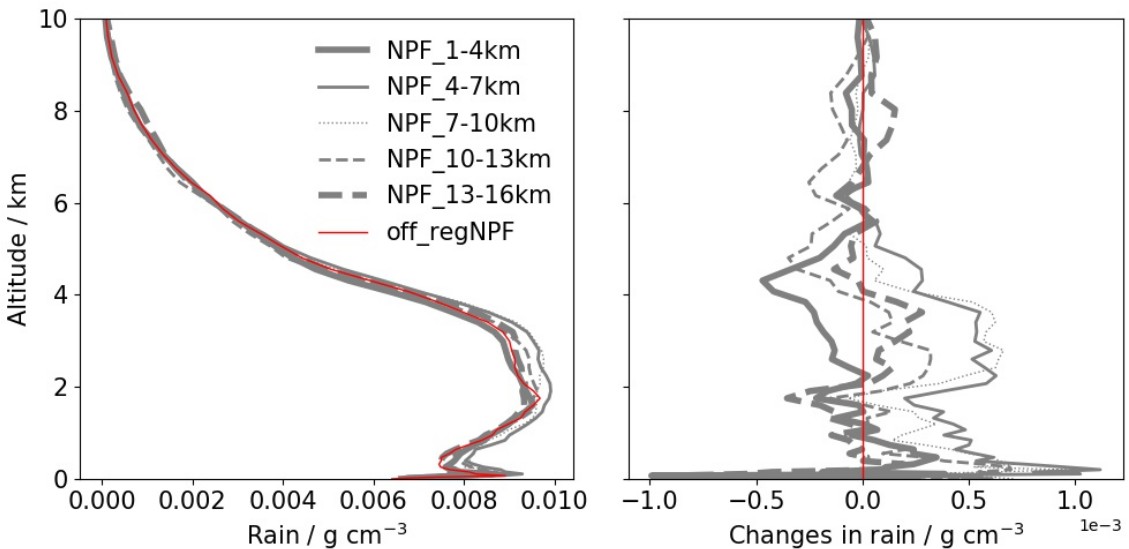

**Figure A11.** Regional domain averaged profiles of rain water contant in the simulations NPF_1-4km, NPF_4-7km, NPF_7-10km, NPF_10-13km, NPF_13-16km, and off_regNPF (left), and the differences between NPF switched on and off in the regional domain (NPF_Xkm − off_regNPF; right).





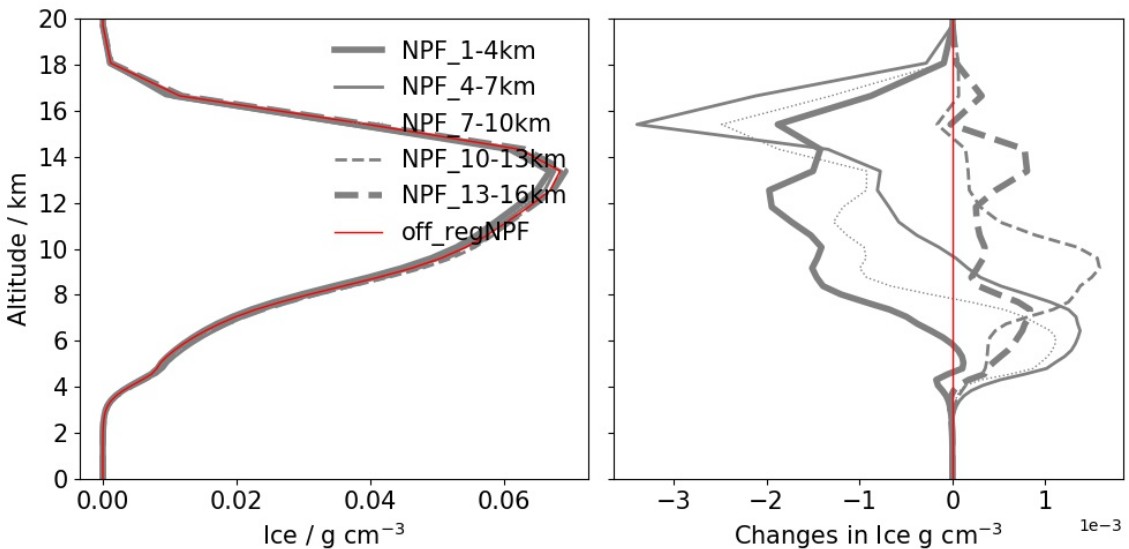

**Figure A12.** Regional domain averaged profiles of ice concentrations in the simulations NPF_1-4km, NPF_4-7km, NPF_7-10km, NPF_10-13km, NPF_13-16km, and off_regNPF (left), and the differences between NPF switched on and off in the regional domain (NPF_Xkm − off_regNPF; right).

*Author contributions.* XW, KSC, HG and DPG designed and led this research. The regional configuration of UM-UKCA was provided by HG and DPG, and HG also provided the codes for pure biogenic nucleation. MOA calculated and corrected the CPC and UHSAS data from ACRIDICO-CHUVA campaign. KSC helped make the schematic diagram (Fig. 14) in discussion and conclusion section. XW ran the model simulations, analysed the model results, and wrote the paper with insights, comments, and edits from KSC, HG and DPG.

*Competing interests.* At least one of the (co-)authors is a member of the editorial board of Atmospheric Chemistry and Physics.

*Acknowledgements.* Our research was funded by the Marie Skłodowska-Curie and was with the CLOUD-MOTION project (grant 764991). KSC acknowledges support from Natural Environment Research Council (NERC) grant for The Aerosol-Cloud Uncertainty REduction project (A-CURE) program (grant NE/P013406/1). DPG was supported by the NERC national capability grant for The North Atlantic Climate
System Integrated Study (ACSIS) program (grant NE/N018001/1) via NCAS, and by the ADVANCE (Aerosol-cloud-climate interactions derived from Degassing VolcANiC Eruptions; NE/T006897/1) program. HG acknowledges support from the NASA ROSES program under





grant number 80NSSC19K0949. MOA is supported by Max Planck Society. The aircraft measurements were from the ACRIDICON-CHUVA campaign. We acknowledge the UK Met Office for providing the support for UM-UKCA-CASIM and Monsoon Superco(o)mputing Node to run the simulations and the JASMIN team with whose platform we processed our modelled data. We thank Ananth Ranjithkumar for

providing the Fortran codes to output nucleation rate and condensation sink diagnostics. We also thank the rest of the members in the aerosol group in the Institute for Climate and Atmospheric Science for the discussions and advice.





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
