# Peer review of "Contribution of regional aerosol nucleation to low-level CCN in an Amazonian deep convective environment: Results from a regionally nested global model"

_Atmospheric Chemistry and Physics, 2022_

## Author Comment (AC1)

**Review ACP-2022-705**

**First review**

We thank the reviewer for his/her useful comments and suggestions which have helped improve the manuscript. The reviewer's comments are listed below in black and our responses are the text in blue. New sentences/paragraphs are added to the manuscript and they are the italic fonts in green.

Contribution of regional aerosol nucleation to low-level CCN in an Amazonian deep convective environment: Results from a regionally nested global model, by Xuemei Wang et al.

**General Comment:**

This well-designed study aims to investigate the new particle formation related to the deep convection transport and the horizontal advection from neighbors' regions by using the HadGEM3 climate model nest with a regional domain over Amazonia. Combining high resolution with resolved convection with the GLOMAP-mode aerosol scheme and the global model with raw resolution allows for testing different hypotheses. The findings are interesting and corroborate empirical data from the ACRIDICON-CHUVA experiment. Furthermore, results allow discriminating of the regional/deep convection and the long-range transport contribution to the aerosol profile in Amazonas.

I would recommend the authors revise the new literature on this subject; for instance, the three recent studies from Bardakov et al. on Tellus, JGR, and James (https://doi.org/10.1080/16000889.2021.1979856,

https://doi.org/10.1029/2022JD037265, and https://doi.org/10.1029/2019MS001931). These studies, using large eddy simulation, were able to present and quantify the convective transport, chemical reactions, and new particle formation in detail. As these models with 100 m resolution are much more appropriate to describe the updrafts and downdrafts, these studies provide a detailed description of the aerosol-deep convection interaction. I recommend authors read/refer to these studies and consider what is new in the present study.

We thank the reviewer for pointing us to these very relevant papers. We add the following sentences to the manuscript.

- In line 35 'and isoprene (Schulz et al., 2018). While the oxidation products of isoprene do contribute to NPF in Amazonian upper troposphere (Schulz et al., 2018), isoprene is also found to suppress NPF from monoterpenes (Heinritzi et al., 2020)'
- In line 52 'The upward transport of insoluble biogenic precursor gas has been found in the experimental study of Kulmala et al. (2006), and the transport process may significantly affect particle nucleation in the upper troposphere. The vertical upward transport efficiency has been further shown to be positively related to biogenic vapour volatility and negatively related to NOx abundance using 100-m resolution large-eddy simulations (LES), which can resolve the

convection and eddies that are important for deep convective transport (Bardakov et al., 2020, 2021, 2022). Upward transport within a relatively short time and domain from the boundary layer to the mid-troposphere was also shown to be efficient in the LES study Bardakov et al. (2022), and it indicates the potential important role of deep convection in transporting precursor gases to the free troposphere and upper troposphere. LES studies found that alphapinene, as a subset of monoterpenes, could be rapidly transported from the surface to above 10 km in altitude and the transport efficiency of isoprene was largely suppressed by NOx due to loss from oxidation (Bardakov et al., 2020, 2021). Bardakov et al. (2022) also showed that downward transport from 7-11 km altitude to the boundary layer was very weak within 2 hours during convection.'

Another comment is related to the deep convection described by the regional model; convection is too deep, deeper than observed. The typical height of deep convective cloud in this region, at this season, is around 9 km (observed by radar – rain drops) and 14 km of cloud top (lidar). Therefore, 20 km, as shown in the figures, is out of the reality of the convective system. It is higher than tropopause (around 16 km).

Yes, the regional model does sometimes produce deeper than real-world convection. Very deep convection is rare but also occurs occasionally in Amazonia (Dodson et al., 2018). Therefore, we agree that the explicit convection is not perfect, but it allows us to investigate the vertical transport of monoterpenes and aerosols.

In general, we expect that the excessive depth of the clouds should not strongly affect the conclusions that deep convective clouds do not transport many aerosols from the UT to the boundary layer, because the convection in the model generates a fairly good and realistic profiles at lower altitudes, which are more important for downward transport to the boundary layer. The extra particles at higher altitudes (> 12 km) are not likely to be transported into the boundary layer. We have added it to the manuscript at line 369.

'Almost all simulations show peak concentrations of ND>20nm and ND>90nm at altitudes higher than the observations. It is possible that the higher altitudes of the peaks in the model are related to relatively deeper convection that transports precursor gases and particles upwards. However, it should also be noted that there are only a few observations to be compared with at high altitudes and it is difficult to conclude what the altitude of the peak was from the observations. Therefore, the modelled results are in general similar to the observations for those altitudes that are important for downward transport to the boundary layer. The modelled extra particles at altitudes higher than 12 km are not relevant to downward transport, and thus, it is also likely that differences in the heights of the peak concentrations would only marginally affect the particle concentration in the boundary layer.'

In addition, the looping simulations show the accumulation and Aitkens mode moving westerly, so above tropopause flow. How do these particles penetrate the stratosphere? Are these features real? The conceptual model presented in the conclusions shows the maximum height as 12 km, well below the layer shown in the results.

Deep clouds are the main reasons for these aerosols to reach high altitude in the model because of the strong upward transport and the associated NPF (and growth). Yes, very deep convection is rare but is real in Amazonia (Dodson et al., 2018). Figure 10 shows that those NPF-induced Aitken and accumulation mode aerosols in the UT are mainly caused by NPF in the UT and a small part of them are formed by NPF at lower altitudes (mainly NPF at 4-10 km) and then transported upward to above 10 km.

We thank the reviewer for commenting on the heights in the conceptual model. The heights in conceptual model in Fig. 14 were set to roughly illustrate the altitude of particles and thus were not matching the simulations in the model. We have adjusted the heights so that they match the model.

The regional model shows an increase of around 100% in the nucleation particle concentration in the lower levels. This is not well discussed in the manuscript. From where do these particles come? Are these particles formed in the boundary layer by monoterpenes oxidation?

Yes, the nucleation mode aerosol particles are formed by NPF from monoterpene oxidation at low levels and a small fraction of them are formed from above followed by downward transport. We also made the following changes to the manuscript.

- at line 478: 'At 1 and 2 km altitude, NPF in the regional domain accounts for nearly 100% of the nucleation mode particle concentration. It again shows that NPF occurs within a short time even at lower altitudes, although Fig. 8 shows that the time- and domain-averaged nucleation mode number concentration below 2 km altitude is less than 200 cm-3, nearly 2 orders of magnitude smaller than at 14 km altitude.'
- at line 569: 'Nucleation mode aerosol number concentration below 2 km in altitude shows positive percentage changes in the simulations NPF\_1-4km, NPF\_4-7km and NPF\_7-10km. The results suggest that the nucleation mode aerosols in the regional domain at lower altitudes (Fig. 9) are primarily formed by NPF at the same heights and a small part of the nucleation mode aerosols are formed by NPF between 3-10 km altitude followed by downward transport.'
- at line 653: 'Although the regional domain NPF accounts for the majority of nucleation mode concentrations in the lower atmosphere, NPF here is much weaker than above 10 km in altitude, and the results are consistent with observational studies that have shown insufficient boundary layer NPF (Zhou et al., 2002; Krejci et al., 2003; Rissler et al., 2006; Spracklen et al., 2006; Andreae et al., 2018; Rizzo et al., 2010; Wimmer et al., 2018).'

I am curious to know the concentrations of Monoterpenes and Ozone employed in the simulations and how they compare with the data measured at ATTO.

We compared the monoterpenes in Bio simulation with the observations at line 285. To make it clearer, we also add the following detailed comparisons (and the tables) to the appendix. Reducing the oxidation rate and increasing the emission rate allows more monoterpenes to be transported to the UT. The observations reported in the corresponding literatures were not between 16-18 September 2014 (the simulation period), but were in the dry season of Amazonia. Therefore, the comparisons between the model and observations show rough differences. Although discrepancies exist

between the model and observations due to different time of the year, averaging method, and simplified chemistry in the model, overall, the O3 in the model (read in from a monthly averaged file) are close to those observed in the upper troposphere and are slightly overestimated the observations at lower altitudes. The adjusted monoterpenes in BioOx simulation overestimate the observations below 2.5 km by factors of around 1-9. The BioOxEm simulation overestimates monoterpenes by a factor of 4-7 and up to a factor of 39 in the model, but such overestimations are reduced as altitude increases until 2.5 km where observations were available.

- At line 755:'As alpha-pinene contributes to around 50% of the monoterpene, we doubled the observed alpha-pinene to estimate the monoterpene in Table A1. The observed monoterpene concentration at the surface is from Yáñez-Serrano et al. (2018), and 80 m and 155 m are from Zannoni et al. (2020). The observed monoterpene concentration at 1-2.5km is from Kuhn et al. (2007). In the simulation with the original default biogenic nucleation mechanism, it shows that monoterpenes are overestimated at altitudes lower than 1 km but underestimated above 2.5 km. The adjusted monoterpenes in BioOx simulation overestimates monoterpenes by a factor of 4-7 and up to a factor of 39 in the model, but such overestimations are reduced as altitude increases until 2.5 km where observations were available.'
- Because BioOxEm has high monoterpene concentrations, we additionally updated Fig. 9 by including extra simulations which switch off NPF based on BioOxCCS, because BioOxCCS simulation reproduces monoterpene concentrations fairly well. We also added an additional paragraph at line 516 to analyse the extra simulations in Fig. 9.
  - At line 516 'Compared to BioOxEmCCS, BioOxCCS simulation shows a relatively reduced dependence of nucleation and Aitken mode number concentrations on NPF in the regional model at almost all altitudes. Thus, the contribution of NPF in the global model to nucleation mode particles becomes greater in BioOxCCS simulation. However, the percentage contribution of NPF in the global model in BioOxCCS simulation to Aitken mode concentrations is smaller than BioOxEmCCS simulation. The smaller percentage in BioOxCCS simulation is related to the suppressed growth from less monoterpene emissions compared to BioOxEmCCS simulation. The percentage contributions of both global and regional NPF to Aitken mode concentrations are smaller compared to BioOxEmCCS simulation because of generally weaker NPF and thus the primary emission represents a greater percentage. For accumulation mode, the percentage contributions of NPF in the regional model of BioOxCCS simulation are similar to the BioOxEmCCS simulation, except for 14 km altitude where NPF in the regional model causes an almost 100% reduction of accumulation mode concentration. It means that with less monoterpene emissions, switching on NPF in the regional model will quickly deplete condensable gases for particle growth at 14 km altitude. Similar to Aitken mode, primary emissions also have a greater percentage concentration to the accumulation mode concentration at 1 and 2 km altitude in BioOxCCS than BioOxEmCCS simulation. At lower altitudes. NPF in the regional model accounts for 1.5% of the total Aitken and accumulation mode particle concentrations at 2 km altitude and 0.2% at 1

km altitude in the BioOxCCS simulation, compared to 3.4% at 2 km and 1.5% at 1 km in altitude for the BioOxEmCCS simulation. The contribution of total Aitken and accumulation mode concentration in the lowest 2 km altitude from the NPF in the global model is between 58% and 65% in BioOxCCS and between 76% and 81% in BioOxEmCCS simulation.'

| Altitude              | monoterpene   | monoterpene at   | monoterpene     | monoterpene   |
|-----------------------|---------------|------------------|-----------------|---------------|
|                       | at ~24 m in   | 80 m in ary      | at 150 m in dry | at 1-2.5 km   |
|                       | Oct 2014      | season in 2017   | season in 2017  |               |
| Monoterpene from      | 0.8 ppbv from | 0.33 ppbv from   | 0.2 ppbv from   | 0.1-0.4 ppbv  |
| observed alpha-pinene | ATTO tower    | ATTO tower       | ATTO tower      | aircraft      |
| Averaged model result | 1.6 ppbv at   | 0.7 ppbv at 75 m | 0.3 ppbv at 155 | 0.019 ppbv at |
| (Bio simulation)      | the surface   |                  | m               | 1-2.5 km      |
| Averaged model result | 4.2 ppbv at   | 2.8 ppbv at 75 m | 1.8 ppbv at 155 | 0.49 ppbv at  |
| (BioOx simulation)    | the surface   |                  | m               | 1-2.5 km      |
| Averaged model result | 37.1 ppbv at  | 23.7 ppbv at 75  | 15.1 ppbv at    | 3.94 ppbv at  |
| (BioOxEm simulation)  | the surface   | m                | 155 m           | 1-2.5 km      |
|                       |               |                  |                 |               |

**Appendix Table A1**

• The following texts are added to line 763: 'O3 in the model is read in from monthly mean ancillary files rather than being calculated online. Table A2 shows the domain averaged O3 mixing ratios from the ancillary file. The observed O3 at 24 m, 53 m, and 79 m are from Andreae et al. (2015), and 11-13.5 km is from Andreae et al. (2018). O3 at lower altitude is overestimated by the model but is about the same magnitude at 11-13.5 km.'

**Appendix A2**

| Altitude | O3 at 24 m     | O3 at 53 m       | O3 at 79 m        | O3 at 11-13.5 km       |
|----------|----------------|------------------|-------------------|------------------------|
| Observed | 2-9ppbv (ATTO) | 5-11 ppbv (ATTO) | 6-12 ppbv (ATTO)  | 25-100 ppbv (aircraft) |
| Model    | 17 ppbv (20 m) | 18 ppbv (53 m)   | 18.9 ppbv (100 m) | 48-52 ppbv             |

Finally, the simulations BioOxEmCCS used in the main simulations showed a peak well above the measured in ACRIDICON-CHUVA (12 km, against 14 km) and with much less middle levels concentration (around twice). What is the effect of these differences in the results?

The discrepancies between the BioOxEmCCS simulation and the observations are likely because we have limited number of aircraft observations to compare with in the free and upper troposphere. Therefore, whether the peak at 14 km is robust is unknown yet. The overestimates at mid-levels would, if anything, increase the supply of Aitken mode to the BL in the model versus the real world.

• We add the following texts to the manuscript to discuss such features at line 359 'which would increase the supply of aerosol particles to the boundary layer in the model versus the real world' and at line 369 'Almost all simulations show peak concentrations of ND>20nm and ND>90nm at altitudes higher than the observations. It is possible that the higher altitudes of the peaks in the model are related to relatively deeper convection that transports precursor gases and particles upwards. However, it should also be noted that there are only a few observations to be compared with at high altitudes and it is difficult to conclude what the altitude of the peak was from the observations. Therefore, the modelled results are in general similar to the observations for those altitudes that are important for downward transport to the boundary layer. The modelled extra particles at altitudes higher than 12 km are not relevant to downward transport, and thus, it is also likely that differences in the heights of the peak concentrations would only marginally affect the particle concentration in the boundary layer.'

Minors comments:

• Line 155 - Discuss the limitations of 4 km resolution in representing the deep convective processes and the grey zone issues.

We agree that 4 km resolution is not able to resolve all the convective processes. We have another simulation at 1.5 km resolution in a similar domain and it showed similar profiles of the particles. Therefore, we briefly describe the simulation at 1.5 km resolution, and add some additional literature discussions (*Li et al. (2018) and Ryu et al. (2022*)) to the limitations section.

- The following texts were added to line 167 '4 km resolution can only resolve part of the convection which may limit our understandings of the efficiency of vertical transport because the convective up- and downdrafts may not be fully resolved. Despite this, it allows us to conduct the simulations and investigate the processes that occur in a relatively large region without high computational cost. Li et al. (2018) and Ryu et al. (2022) have shown that coarser resolutions may generally cause an earlier onset of convection, but the general horizontal distribution of clouds should not be strongly affected. In a test simulation with 1.5 km resolution and a similar domain, we found that the Aitken and accumulation mode concentration profiles were similar to the aerosol in the same region at 4 km resolution. Therefore, 4 km is sufficient to investigate this topic.'
- Line 210 Please specify the profiles of gas assimilated and explain if they are fixed, or the chemical processes consume them. Convection brings ozone into the boundary layer, as mentioned in the text. Why does this process not modulate ozone concentration in the boundary layer? Isoprene has around ten times more concentration than monoterpenes. Why is isoprene not included in the chemical process?

The oxidants are read in from monthly averaged ancillary files. They are prescribed and not affected by chemical processes or convection.

• We add the following texts to indicate the oxidants fields to line 229 'with prescribed oxidant fields (OH, O3, H2O2, HO2, and NO3)' and to line 231 'The fixed oxidants are not affected by chemical reaction or convection.'

Ozone is not transported in our model, but the study of Gerken et al. (2015) investigated ozone transport which is similar to the aerosol vertical transport in our study.

• We realised that the phrasing in discussion caused confusion, so we changed the sentence to line 681 'The extent of aerosol vertical transport in our simulations is similar to the study of Gerken et al. (2015) who showed that O3 from 2-7 km altitude may enter the boundary layer during convective storms occurring during GoAmazon2014/5.'

We agree that isoprene is very important. However, it was not included in the offline chemistry.

- We added this statement to the method at line 234 'but the simplified chemistry scheme does not include isoprene and the related chemistry'.
- We also add a sentence to the limitation (line 739) to state the missing isoprene may cause some uncertainty: 'Another limitation is that isoprene emission and the corresponding chemistry were not included with the simplified offline chemistry scheme, and thus it may to some extent limit our understanding of NPF and particle formation in this region.'
- Line 235 The average maximum rain rate seems very high (118 mm/hr). Convection in the model usually occurs at 1100 LST and rainfall at 1300 LST. However, convection in Central Amazonas occurs later, and precipitation occurs around 1600 LST. Please comment on how do this early convection impact the results?

Yes, 118 mm/hr is quite an unreal number. We estimated the number 118 mm/hr from instantaneous rain rate values (sampled every 3 hours), but over longer averaging periods, it would be lower. It does not seem a good way to show the rain that occurs in the model. It has been removed from the manuscript.

The early convection is expected to transport monoterpenes to the UT at an earlier time, therefore, monoterpenes can be oxidised earlier and contribute to an earlier NPF as well as particle growth. But because the modelled profiles are close to the observations, the early convection is not strongly affecting the conclusions.

- To specify the effect of early convection, we add the following sentences to line 263 'The onset of the convection in the model is earlier than observed in Amazonia. This early onset may result in an earlier occurrence of new particle formation in the UT. However, because we tested several nucleation mechanisms and hence explored a range of different nucleation rates in order to match the observations, the early convection should not significantly affect our conclusions.'
- Line 255 "We do not increase the oxidation rates because they will drive the simulations away from the observations by producing too few aerosols in the UT". Should this effect happen because isoprene has not been considered in the simulation? When enough oxidation is available, I see no reason for the isoprene not to be considered.

Including isoprene would not affect the conclusions. We have a few other simulations with coupled chemistry (interactive oxidant fields) and isoprene (emission and chemistry) and they still show that a slower oxidation rate of monoterpenes will produce more particles via NPF. Including isoprene would be ideal in this set of simulations.

However, isoprene and the associated reactions are not included in our main set of simulations with offline chemistry .

- Therefore, we add it to the limitations to line 282 'We have some other simulations that used isoprene emission and chemistry, and interactive oxidant fields in UKCA. They also showed that faster oxidation rates of monoterpenes tend to produce fewer aerosol particles.'
- Line 410 Figure 10 legends are wrong (NPF altitude range /km);

We thank the reviewer for spotting the error. The figure is now updated with the corrected x-axis labels

• Line 520 – Figure 11, The vertical motion inside the clouds appears very low to be the core of deep convection.

The updraft speed look small because of the colorbar scale. When the maximum and minimum values are not manually set, the vertical velocities reflect the deep convection better as are shown in the figures below.

Additionally, the vertical slices represent only part of the clouds that exist at 1.64 S, so they may not represent all the clouds cores, thereby they may show up as relatively low vertical velocity.

---

## Author Response (AR2)

We thank the editor for the comments. The comments are listed below in black and our responses are the text in blue. New sentences/paragraphs are added to the manuscript and they are the italic fonts in green. Line numbers of new texts are listed according to the manuscript with tracked changes.

"Recent measurements during CAFÉ-Brazil clarify the importance of isoprene in the formation of nucleation particles in the UT. For this reason, quantifying the processes described in the simulations could be biased by considering only the monoterpenes. Hence, I recommend highlighting in the conclusion that this study does not consider isoprene."

Reply: We agree that the missing isoprene chemistry and emission may affect the representation of particle concentration profiles in the results. To specify this, we added the following to the limitations at line 743. The new sentences are as follows:

- *'The oxidation product of isoprene has been found to contribute to around 20 % of the total secondary organic aerosol mass (Schulz et al., 2018). Yee et al. (2020) also stated the importance of isoprene in forming preindustrial aerosol sulfate in the environments with high isoprene emissions. Thus, our results on model-observation comparisons in Sect. 3.1 may not well represent the particle compositions in Amazonian environment because we only considered monoterpenes'*

In addition, the limitation in the 4 km resolution simulation does not allow a precise description of the clouds' macro and microphysical structure. Therefore, the results should be interpreted only qualitatively."

Reply: Yes, 4 km resolution cannot fully represent the cloud properties. We added new texts at line 732 and 735 to the limitations to clarify this fact. The newly added sentences are as follows:

- line 732: *'The relatively coarse 4 km resolution cannot fully resolve all the cloud-related processes and thus, 4 km resolution may not represent the full details of the cloud macro structure and microphysics'*
- line 735: *'Then, our results that showed the number of vertically transported aerosols can only be interpreted qualitatively.'*

In addition to the changes above, we also updated the colours in Fig. 1, 2, 8, 10, 11 and 12 to make the plots easier to observe under Coblis — Color Blindness Simulator.